# SPQR: Controlling Q-ensemble Independence with Spiked Random Model for Reinforcement Learning

**Dohyeok Lee**[1], **Seungyub Han**[1,2], **Taehyun Cho**[1], **Jungwoo Lee**[1,*]
[1]Seoul National University, [2]Hodoo AI Labs
{dohyeoklee,seungyubhan,talium,junglee}@snu.ac.kr

## Abstract

Alleviating overestimation bias is a critical challenge for deep reinforcement learning to achieve successful performance on more complex tasks or offline datasets containing out-of-distribution data. In order to overcome overestimation bias, ensemble methods for Q-learning have been investigated to exploit the diversity of multiple Q-functions. Since network initialization has been the predominant approach to promote diversity in Q-functions, heuristically designed diversity injection methods have been studied in the literature. However, previous studies have not attempted to approach guaranteed independence over an ensemble from a theoretical perspective. By introducing a novel regularization loss for *Q-ensemble independence* based on random matrix theory, we propose *spiked Wishart Q-ensemble independence regularization* (SPQR) for reinforcement learning. Specifically, we modify the intractable hypothesis testing criterion for the Q-ensemble independence into a tractable KL divergence between the spectral distribution of the Q-ensemble and the target Wigner's semicircle distribution. We implement SPQR in several online and offline ensemble Q-learning algorithms. In the experiments, SPQR outperforms the baseline algorithms in both online and offline RL benchmarks.

## 1 Introduction

Reinforcement learning (RL), especially *deep reinforcement learning* (DRL) that contains high capacity function approximators such as deep neural networks, has shown considerable success in complex sequential decision making problems including robotics [1], video games [28], and strategy games [12]. Despite these promising results, modern DRL algorithms consistently suffer from sample-inefficiency and overestimation bias.

Vanilla off-policy Q-learning faces the overestimation issue in both online and offline reinforcement learning. The overestimation problem is specifically worse for out-of-distribution (OOD) state-action pairs. In order to address the overestimation problem, many existing methods utilize ensemble Q-learning. Ensemble Q-learning introduces multiple Q-networks to compute a Bellman target value to benefit from the diversity of their Q-values. Double Q-learning [15] proposes clipped Q-learning as the first method to adopt an ensemble to DRL. Along with double Q-learning, Ensemble DQN [3], Maxmin Q-learning [22], *randomized ensembled double Q-learning* (REDQ) [6], and SUNRISE [24] are proposed as efficient ensemble Q-learning algorithms on online RL tasks. Clipped SAC [2] is proposed for offline RL tasks to alleviate the overestimation bias of Q-values from OOD data. Despite the objective of ensemble Q-learning, which enables RL agents to access diverse Q-values for each state-action, existing ensemble Q-learning algorithms have only focused on the computing Bellman target over an ensemble. MED-RL [31] and EDAC [2] propose a diversifying loss for online and offline reinforcement learning, such as the Gini coefficient and the variance of the Q-value on

---

*Corresponding author

37th Conference on Neural Information Processing Systems (NeurIPS 2023).

OOD data. However, these methods have a limitation in that the proposed losses do not guarantee independence from a theoretical perspective, but diversify Q-functions from heuristics and empirical studies although they have an i.i.d assumption. Most ensemble Q-learning algorithms assume that bias follows a uniform and independent distribution, which may not be true when using deep neural network approximation and a shared Bellman target. Hence we need to reinvestigate the notion of *Q-ensemble independence* more rigorously through a random matrix theory. The random matrix theory studies the statistical behavior of a random matrix and has a wide range of applications in diverse fields such as physics, engineering, economics, and machine learning as shown in [5, 25, 29].

In this paper, we propose *spiked Wishart Q-ensemble independence regularization* (SPQR), a tractable Q-ensemble independence regularization method based on the random matrix theory and a spiked random model perspective. SPQR is a regularization method that theoretically guarantees to improve the independence of a Q-ensemble during training. SPQR penalizes the KL divergence between the eigenvalue distribution of the Q-ensemble and an ideal independent ensemble. If a Q-ensemble becomes closer to the ideal independent ensemble, it satisfies the independence assumption of ensemble Q-learning. To show how effective SPQR is, we combine SPQR with various algorithms and online/offline RL tasks. SPQR outperforms baseline algorithms in MuJoCo, D4RL Gym, Franka Kitchen, and Antmaze. To summarize, our contributions are as follows:

1. We propose a tractable loss function for Q-ensemble independence without any prior assumption of Q-function distribution by adopting random matrix theory and a spiked random model for the first time in DRL algorithms.

2. It is empirically shown that the universality and performance gain of SPQR for various algorithms and tasks. SPQR is applied to SAC-Ens, REDQ, SAC-Min, CQL, and EDAC and evaluated in various complex and sparse tasks in online and offline RL, MuJoCo, D4RL Gym, Franka Kitchen, and Antmaze. To ensure our reproducibility and fair comparisons, we prodive our source code.[1]

## 2   Related Works

**Ensemble Reinforcement Learning.**   Ensemble DQN [3] is the first method to adopt ensemble Q-learning for DRL. Ensemble DQN aims to reduce overestimation bias by computing a Bellman target as an average Q-value in the ensemble. Maxmin Q-learning [22] computes a Bellman target as a minimum Q-value over the Q-ensemble and shows significant performance enhancement based on the Atari benchmark. REDQ [6] constructs sample efficient ensemble Q-learning algorithm based on Maxmin Q-learning and SAC with a probabilistic ensemble by selecting a subset over Q-ensemble. Since REDQ combines the *update-to-data* (UTD) ratio, which is an orthogonal method with ensemble subset minimization, REDQ achieves state-of-the-art performance in the online MuJoCo environment. For offline reinforcement learning tasks, clipped Q-learning [2], denoted as SAC-Min is proposed, which computes the Bellman target as in Maxmin Q-learning. Diversification methods are proposed to improve the performance of ensemble Q-learning. MED-RL [31] optimizes the Gini coefficient to diversify an ensemble, and EDAC [2] optimizes the variance of the Q-value on OOD data to diversify an ensemble from the OOD data. SPQR proposes a Q-ensemble independence loss from a theoretical perspective, based on a spiked random model and random matrix theory to make ensemble Q-networks as independent as possible. In addition, *conservative Q-learning* (CQL) is proposed by [21], which penalizes the Q-values of OOD actions and performs well on numerous offline datasets.

**Spiked Random Model.**   Signal recovery for a generative model using a spiked random model is analyzed by [8]. SPQR analyzes Q-ensemble independence for ensemble Q-learning in a spiked random model perspective. A universal data-driven optimal likelihood hypothesis test with linear spectral statistics (LSS) for a spiked random model as a central limit theorem for test statistics is given by [7, 18]. SPQR proposes a tractable universal data-driven test $T(\lambda)$ to regularize the Q-ensemble independence.

---

[1]Code implementation is available at: `https://github.com/dohyeoklee/SPQR`

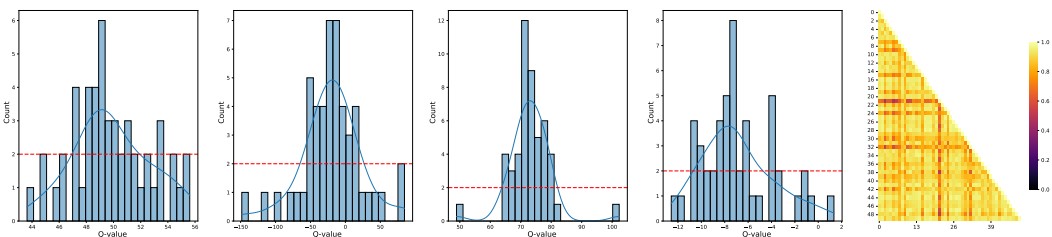

Figure 1: **Left 1–4:** Histogram visualization of Q-value distribution over networks in the ensemble. Each plot represents a Q-value histogram for one state-action data. The X-axis represents the Q-value of each Q-network and the Y-axis represents the number of Q-networks in the histogram bin. The red horizontal line represents a uniform distribution and the blue solid line represents kernel density estimation for a given histogram. **Rightmost:** Heatmap visualization of the Pearson correlation coefficient matrix between each Q-network in the ensemble. Detailed values and explanations are given in Appendix E.

## 3 Backgrounds

In this section, we describe ensemble Q-learning algorithms and visualize an empirical study of their limitations. In order to overcome its limitations, we introduce the random matrix theory and a spiked random model to formulate the notion of Q-ensemble independence from a theoretical perspective.

We consider a *Markov decision process* (MDP), defined by a tuple $(\mathcal{S}, \mathcal{A}, P, r, \gamma)$, where $\mathcal{S}$ is a finite state space, $\mathcal{A}$ is a finite action space, $P : \mathcal{S} \times \mathcal{A} \times \mathcal{S} \to [0, 1]$ is the state-action transition probability, $r : \mathcal{S} \times \mathcal{A} \to \mathbb{R}$ is the reward function, and $\gamma \in [0, 1)$ is the discount factor. We define a state-action value function, called Q-function, $Q_\phi : \mathcal{S} \times \mathcal{A} \to \mathbb{R}$, represents an expectation of the discounted sum of rewards for the trajectory started from $s, a$, given by $\mathbb{E}_{s_t, a_t}[\sum_{t=0}^{\infty} \gamma^t r(s_t, a_t)]$.

### 3.1 Ensemble Q-learning

We first define bias $e_{sa}^i = Q^*(s, a) - Q_i(s, a)$, where $Q^*(s, a)$ is the optimal Q-value given for $(s, a)$ and $Q_i(s, a)$ is the Q-value of the $i$-th networks in ensemble. Ensemble Q-learning assumes that the bias follows a zero mean, uniform, and independent distribution, i.e., $e_{sa}^i \sim Unif(-\tau, \tau)$ and is independent with respect to $i$, although REDQ does not assume uniform distribution. Given $N$ Q-functions in the ensemble, there are three typical methods to compute a Bellman target: Ensemble DQN, Maxmin Q-learning, and REDQ.

**Ensemble DQN.** Ensemble DQN computes the Bellman target as the ensemble average Q-value. The bias and variance of the bias monotonically decrease as the ensemble size $N$ increases, and they become 0 as $N$ goes to infinity. We implement the SAC-style of Ensemble DQN, denoted as SAC-Ens.

**Maxmin Q-learning.** The Bellman target of Maxmin is to select the minimum Q-value within an ensemble, which can control the level of overestimation with a finite number of Q-functions. A modified version called SAC-Min is introduced specifically for offline RL tasks.

**REDQ.** REDQ computes the Bellman target as the minimum Q-value in a subset of the ensemble to become a sample-efficient algorithm.

Detailed explanations for Bellman targets and the evaluation function of each algorithm, denoted as $Ens_{tar}$ and $Ens_{eval}$, are described in Appendix C.

To confirm the validity of uniform and independent assumptions of ensemble Q-learning, we plot Q-value distribution in Figure 1 and test whether each Q-distribution is uniform using the $\chi^2$ test. We train 50 Q-networks 3M timesteps by the SAC-Min algorithm in the D4RL hopper-random dataset and evaluate it by the hopper-full-replay dataset. We also perform the $\chi^2$ test for independence in Table 1. Out of 1000 batch data, only 35.8% can accept that the distribution is uniform and 30.4% can accept independence. Thus, $e_{sa}^i \sim Unif(-\tau, \tau)$ and i.i.d. assumption is empirically inaccurate and it is necessary to introduce the notion of Q-ensemble independence. More details on the $\chi^2$ test are given in Appendix E.

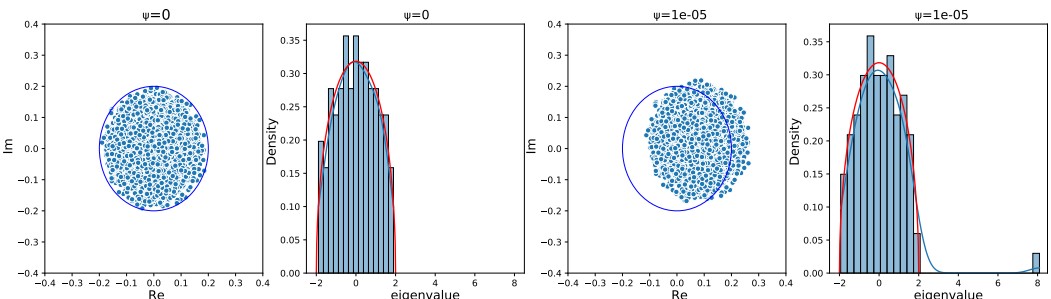

Figure 2: Eigenvalue plot for the spiked Wishart model with Wigner's semicircle law. For effective visualization of the spiked model, we use a complex hermitian random matrix, GUE. Blue dots represent each data in a complex plane. The red line represents Wigner's semicircle distribution. The blue line in the histogram represents kernel density estimation. **Left 1–2:** Perturbation power is $\psi = 0$. Eigenvalue distribution follows Wigner's semicircle law. **Right 3–4:** Perturbation power is $\psi = 10^{-5}$. Eigenvalue distribution almost follows Wigner's semicircle law except for the largest eigenvalue, interpreted as a spike.

As [30] also points out, the early collapse of Q-values, which means that the Q-values in the ensemble become almost identical, occurs during the training. We empirically visualize a strong correlation among Q-networks in Figure 1. Although the main advantage of ensemble Q-learning is based on diverse Q-functions, network initialization is the only process to maintain diversity over an ensemble since the Bellman target is identical across all networks without diversification and independence regularization methods.

### 3.2 Random Matrix Theory

Random matrix theory studies the probabilistic properties of large matrices with random entries. Dyson [10] constructs some categories of the random matrix as a real symmetric matrix as *Gaussian orthogonal ensemble* (GOE), complex hermitian matrix as *Gaussian unitary ensemble* (GUE), and quaternion self-dual matrix as *Gaussian symplectic ensemble* (GSE), called *Dyson's classification*. In this paper, we restrict our interest to matrices in GOE.

**Definition 3.1** (Gaussian Orthogonal Ensemble, [26]). Let random matrix $W \in R^{N \times N}$, its entities random variables $w_i$, $P[W] = P(w_1, \ldots, w_{N^2})$ to be a joint probability density function of $w_i$. Define the set of $W$ as GOE when these properties hold:

(1) $w_i$ are independent with respect to $i$.
(2) $W$ is a real symmetric matrix.
(3) $W$ is rotational invariance, i.e., $P[W]dW = P[W']dW'$ if $W' = UWU^{-1}$, $U$ is orthogonal.

GOE is the most natural way to model random matrices in various applications, including machine learning. In simplified terms, we can consider GOE as a Wigner matrix, where entries are independent, symmetric, and follow Gaussian distribution. One of the main areas in random matrix theory is how eigenvalues or singular values of a large random matrix are distributed. More specifically, we only consider limiting properties of *empirical spectral density* (ESD), which is defined as, for random matrix $W \in \mathbb{R}^{N \times N}$ and its eigenvalues $\lambda_i, i = 1, \ldots, N$,

$$p_W(\lambda) = \frac{1}{N} \sum_{i=1}^{N} \delta(\lambda - \lambda_i) \tag{1}$$

The limits of the empirical spectral density of the GOE, called the *Wigner's semicircle law*, are given by [34]. For GOE $\hat{W} \in \mathbb{R}^{N \times N}$ with variance of entries $\sigma^2$, $\lambda_1, \ldots, \lambda_N$ to be a eigenvalue of $W = \frac{1}{\sqrt{N}} \hat{W}$ and empirical spectral distribution $p_W = \frac{1}{N} \sum_{j=1}^{N} \delta(\lambda - \lambda_j)$, then $p_W$ converges to $p_{sc}$ weakly almost surely, where

$$p_{sc}(\lambda) = \frac{1}{2\pi\sigma^2} \sqrt{4\sigma^2 - \lambda^2} \mathbb{1}_{[-2\sigma, 2\sigma]} \tag{2}$$

We call $p_{sc}$ as the Wigner's semicircle distribution. Wigner's semicircle law does not rely on the probability density of entries.

### 3.3 Spiked Random Model

Random matrix theory models random data well, but in reality, most data have both informative signals and random components. A spiked random model is a theory about constructing a model of perturbation with a data matrix. The objective of this theory is to separate true signals from perturbated ones with random noise. True signals show *spiked* phenomena that make the largest eigenvalue detach from the bulk of eigenvalues. A spiked model was first introduced by [17] as an application for principal component analysis. We restrict our analysis to a *spiked Wishart model*, the most natural methodology to model a general perturbation. A spiked Wishart model is defined as follows.

**Definition 3.2.** Let $Y \in \mathbb{R}^{N \times M} \sim \mathbb{P}_\psi$, $u, v$ is unknown vector, $Y$ is **spiked Wishart model** when

(1) $Y = \sqrt{\frac{\psi}{N}} uv^T + W$, which is equivalent to sample covariance matrix $\hat{\Sigma} = I_N + \frac{\psi}{N} uu^T$

(2) $u, v$ are i.i.d from zero mean priors $u \sim P_u, v \sim P_v$, support bounded in radius by $K_u, K_v$

(3) $W \sim \mathbb{P}_0$ is a matrix with i.i.d noise entries

(4) $M/N \to \omega$

We will assume the spiked Wishart model with $N = M$, where we can ignore property 4. In Figure 2, we visualize how the spiked Wishart model works in both the data and spectral domain. For a detailed explanation, see Appendix A.

## 4 Q-ensemble Independence Regularization

In this section, we present a method of Q-ensemble independence regularization that is computationally feasible through the utilization of a spiked Wishart model. We bring the concept of ensemble Q-learning to the spiked Wishart model as $Y \sim \mathbb{P}_\psi$ representing learned Q-value by Q-learning algorithm and $W \sim \mathbb{P}_0$ representing the perfectly independent Q-ensemble. If $W$ dominates $Y$, $Y$ follows independent assumptions of ensemble Q-learning. If not, a collapse of Q-value and high correlation occurs, which cannot be benefited by the ensemble method. By integrating with ensemble Q-learning and Q-ensemble independence testing, we propose a tractable Q-ensemble independence loss with gain $\beta$, which is further described in the following section. To help understand the independence and diversity of Q-ensemble more clearly, we mentioned that $\beta$ can be simply considered as a loss gain (weight) for Q-ensemble independence regularization before providing a detailed explanation about SPQR and $\beta$. A higher $\beta$ represents highly-likely-to-independent.

### 4.1 Diversity and Independence of Q-ensemble

To clarify our work, we distinguish between the notions of diversity and independence. Numerous ensemble Q-learning methods improve their performance by minimizing the overestimation bias and the variance of the Bellman target estimate. However, such works assume that the biases $e_{sa}^i$ are zero-mean independent and identically distributed over $i$, although early collapse, a high correlation between Q-ensemble, occurs due to the shared Bellman target. On the other hand, diversification methods such as EDAC develop an ensemble diversification objective to reduce the number of ensemble networks required. In particular, EDAC increases the variance of the ensemble to penalize OOD actions. In our work, we focus on the independence of a Q-ensemble to reduce the effect of correlated behavior resulting from a shared Bellman target. Roughly speaking, diversification can be interpreted as flattening the distribution of a Q-ensemble, while independence regularization can be interpreted as promoting

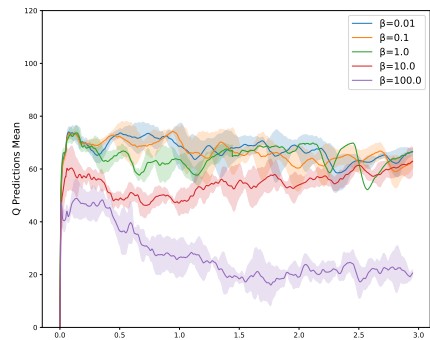

Figure 3: Mean of predicted Q-value of SPQR-SAC-Min with various $\beta$ on hopper-random dataset, averaged over 4 seeds.

an i.i.d. sampling procedure for a given distribution. We verify that our proposed regularization successfully exploits both the conservative behavior and the diversification of a Q-ensemble. The Q-ensemble independence regularization empirically shows a large performance gain by controlling the conservatism, as shown in Figure 3, similar to the performance improvement of the ensemble method caused by the conservatism. For further details, see Appendix E.

## 4.2 Tractable Loss for Q-ensemble Independence

We construct a tractable testing criterion to determine whether a distribution is independent in Theorem 4.1. $D_{KL}(p_\psi(\lambda), p_{sc}(\lambda))$ does not require any prior assumption about $\mathbb{P}_0$ and $\mathbb{P}_\psi$ since in spectral domain, Wigner's semicircle law is universal for all random matrices. Proof of Theorem 4.1 is shown in Appendix B.

**Theorem 4.1.** *Following definition 3.2, as $N \to \infty$ with probability at least $1 - \delta$, following test $T(\lambda)$ is optimal.*

$$T(\lambda) = \mathbb{1}(D_{KL}(p_\psi(\lambda), p_{sc}(\lambda)) \geq \varepsilon) = \begin{cases} H_0 : Y \sim \mathbb{P}_0 \\ H_1 : Y \sim \mathbb{P}_\psi \end{cases}$$

We now describe a practical Q-ensemble independence loss for ensemble Q-learning using the spiked Wishart model and Theorem 4.1. Let $Y$ and $u, v$ for the spiked Wishart model denote a Q-value matrix and an unknown signal. By the test $T(\lambda)$, if $D_{KL}(p_\psi(\lambda), p_{sc}(\lambda)) < \varepsilon$ or is minimized, it is theoretically guaranteed that $Y$ is a random matrix, i.e. a Q-ensemble is independently distributed.

In conclusion, we propose $D_{KL}(p_\psi(\lambda), p_{sc}(\lambda))$ as a Q-ensemble independence regularization loss for ensemble Q-learning, without any prior assumption such as uniform distribution.

## 4.3 Algorithm

As described in Section 3.1, ensemble Q-learning has crucial but inaccurate assumptions about independence. To address the issue of insufficient independence, we propose **SPQR**: spiked Wishart Q-ensemble independence regularization for reinforcement learning, which we describe in Algorithm 1. Using the random matrix theory and a spiked random model lens, SPQR minimizes $D_{KL}(p_\psi(\lambda), p_{sc}(\lambda))$ as a form of regularization loss, denoted as SPQR loss, on the baseline ensemble Q-learning algorithms. $D_{KL}(p_\psi(\lambda), p_{sc}(\lambda))$ penalizes for going too far from an independent Q-ensemble, which means promoting Q-ensemble to follow $\mathbb{P}_0$. We visualize our concept in Figure 4. To compute the SPQR loss using $p_\psi(\lambda)$, we need to construct a symmetric Q-matrix $Y$ filled with Q-values. Then, we normalize the Q-matrix $Y$ with a zero-mean and unit variance to compute the Wigner's semicircle distribution $p_{sc}(\lambda)$ with $\sigma = 1$. Finally, we compute eigenvalues with a numerically stable algorithm by constructing the ESD $p_\psi(\lambda)$ and calculate the KL divergence between $p_\psi(\lambda)$ and $p_{sc}(\lambda)$. The loss function $\mathcal{L}_{SPQR}$ is computed as a expectation of $D_{KL}(p_\psi(\lambda), p_{sc}(\lambda))$ over the batch data. A detailed explanation of implementations and algorithms is shown in Appendix C.

---

**Algorithm 1** SPQR

1: **Input:** $\gamma, \alpha, N, \beta, \tau, |B|$
2: Initialize policy $\theta$, Q-function $\phi_i$, target Q-function $\phi'_i$, replay buffer $\mathcal{D}$
3: **repeat**
4:     $y(r, s') \leftarrow r + \gamma\Big(Ens_{tar}(\phi'_i) - \alpha \log \pi_\theta(a'|s')\Big)$
5:     **for** $i = 1, \ldots, N$ **do**
6:         $\mathcal{L}_{SPQR} = D_{KL}(p_\psi(\lambda), p_{sc}(\lambda))$
7:         $\nabla_{\phi_i} \frac{1}{|B|} \sum (Q_{\phi_i}(s, a) - y(r, s'))^2 + \beta \mathcal{L}_{SPQR}$
8:     **end for**
9:     $\nabla_\theta \frac{1}{|B|} \sum_{s \in B} \Big(Ens_{eval}(\phi_i) - \alpha \log \pi_\theta(\tilde{a}|s)\Big)$
10: **until**

---

**Algorithm 2** SPQR Loss

1: **Input:** Q-network $l_k = \{Q_{\phi_i}(s', a')\}_{i=1}^N$
2: $Y \in \mathbb{R}^{D \times D}$, $D = \lfloor \frac{\sqrt{1+8N}-1}{2} \rfloor$:
3: $Y_{pq} \leftarrow \begin{cases} l_{Dp+q}, & \text{if } p \geq q \\ Y_{qp}, & \text{otherwise} \end{cases}$
4: $Y \leftarrow (Y - \mu_Y)/\sigma_Y$
5: $\{\lambda_i\}_{i=1}^N \leftarrow Eigen(\frac{1}{\sqrt{N}}Y)$
6: $p_{esd}(\lambda) = \frac{1}{N} \sum_i \delta(\lambda - \lambda_i)$
7: $p_{wigner}(\lambda) = \frac{\sqrt{4-\lambda^2}}{2\pi}$
8: $\mathcal{L}_{SPQR} \leftarrow \frac{1}{|B|} \sum_j \sum_i p_{esd}(\lambda_i) \log \frac{p_{esd}(\lambda_i)}{p_{wigner}(\lambda_i)}$
9: **Output:** SPQR loss $\mathcal{L}_{SPQR}$

---

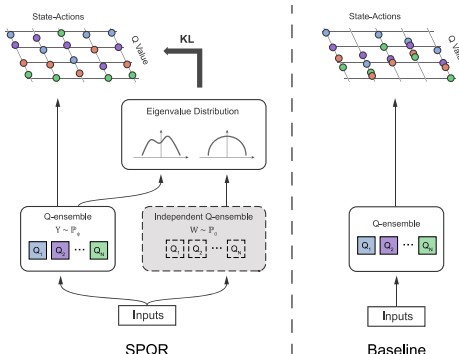

Figure 4: Illustrative example of SPQR.

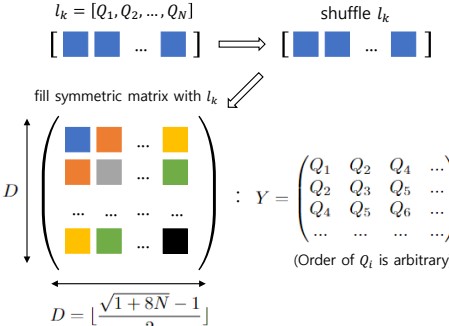

Figure 5: Illustration for building Q-matrix.

## 5 Experimental Results

To reveal the validity and universality of SPQR from algorithms to tasks, we implement SPQR in various ensemble Q-learning algorithms and various online/offline RL tasks. We implement SPQR in SAC-Ens, REDQ, SAC-Min, and CQL, and evaluate them in online MuJoCo Gym tasks, offline D4RL Gym, Franka Kitchen, and Antmaze tasks. We denote the SPQR-implemented version of each ensemble algorithm as SPQR-*, such as SPQR-SAC-Ens. All the following implementations use the same hyperparameter and experimental protocol with a baseline. For fair evaluation, the SPQR code is written on top of each official baseline code, REDQ[1], EDAC[2], and CQL[3]. Implementation of SPQR can be done by only **50** lines and training time only increases by **5**%. We empirically show that SPQR outperforms baseline ensemble Q-learning algorithms in both online and offline RL tasks. See the implementation details in Appendix D.

**Is SPQR really affecting independence?** We perform independence hypothesis testing and visualize the spectral distribution to investigate whether SPQR really affects the independence of the Q-ensemble. First, we analyze the performance gain of SPQR from a spike reduction perspective. We train a Q-ensemble using SAC-Min and SPQR-SAC-Min on the hopper-random dataset and test it on the hopper-full-replay dataset. Figure 6 and Table 1 show the evaluation results for the spike-reducing phenomenon of SPQR, which deviates less from Wigner's semicircle law. Compared to SPQR-SAC-Min with $\beta = 0.1$, SAC-Min has significantly more spikes, in particular, 2434 spikes compared to 2182 spikes over 25600 data. For extreme regularization gain, $\beta = 100$, the spike is drastically reduced to 698, **71**% less than the baseline algorithm. We also plot a spike histogram for another diversification method, EDAC. EDAC increases spikes to **54**% more than SAC-Min, which can be interpreted as SPQR being the orthogonal method with EDAC.

We also demonstrate the $\chi^2$ test with null hypothesis $H_0 : P(Q_i, Q_j) = P(Q_i)P(Q_j)$ and significant level $\alpha = 0.025$ for independence hypothesis testing on the same experimental setting above. Since SPQR promotes being "more" independent, we use the proportion of data that can accept an independent hypothesis as a measure of more or less independence. In Table 1, we report the proportion of trials over 1000 tests that accept the null hypothesis $H_0$, meaning that the Q-ensemble is independent. These empirical studies can be interpreted as SPQR having Q-ensemble independence regularization power since the acceptance rate increases monotonically as $\beta$ increases. Orthogonality between SPQR and EDAC is also found from an independence testing perspective. However, reducing spikes and being more independent does not guarantee optimal performance since there is an optimal conservatism for each task. We can find the proper conservatism by controlling the independence regularization power, $\beta$. More details can be found in Appendix E.

---

[1]`https://github.com/watchernyu/REDQ`
[2]`https://github.com/snu-mllab/EDAC`
[3]`https://github.com/aviralkumar2907/CQL`

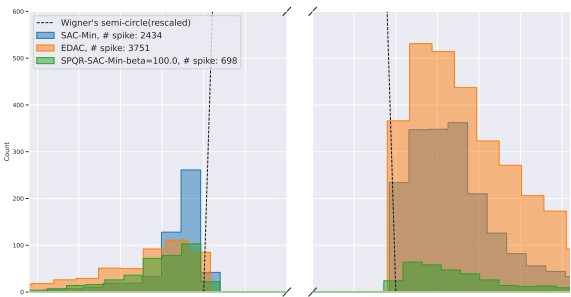

Figure 6: Histogram of spikes for Wigner's semicircle (black line), SAC-Min (blue), EDAC (orange), and SPQR-$\beta = 100.0$ (green). For effective visualization, we plot only the eigenvalues that violate Wigner's semicircle.

Table 1: Number of spikes, independence hypothesis testing acceptance ratio, and performance of SAC-Min, EDAC, and SPQR-$\beta = 0.01, 0.1, 1.0, 10.0, 100.0$. We reproduce EDAC-$\eta = 1.0$. The reported value of EDAC-$\eta = 1.0$ is given in the Section H of [2]

| Algorithm | Spike | Performance | Accept ratio |
|---|---|---|---|
| SAC-Min | 2434 | $31.3 \pm 0.0$ | 30.4% |
| EDAC-$\eta = 1.0$ | 3751 | $10.3 \pm 0.2$ | 0.0% |
| SPQR-$\beta = 0.01$ | 2360 | $33.6 \pm 1.8$ | 59.0% |
| SPQR-$\beta = 0.1$ | 2182 | $\mathbf{35.6 \pm 1.4}$ | 80.4% |
| SPQR-$\beta = 1.0$ | 1844 | $32.6 \pm 0.4$ | 91.4% |
| SPQR-$\beta = 10.0$ | 1603 | $32.7 \pm 0.4$ | 92.3% |
| SPQR-$\beta = 100.0$ | **698** | $32.9 \pm 0.8$ | **97.6%** |

**Online Reinforcement Learning.**    We implement SPQR-SAC-Ens and SPQR-REDQ with UTD ratio 1 and 20, and compare them to SAC-Ens and REDQ to show how SPQR improves performance. We evaluate each algorithm on the OpenAI Gym MuJoCo tasks. Figure 7 shows the training curves for SAC-Ens, REDQ, SPQR-SAC-Ens, and SPQR-REDQ. Since the two main contributions of REDQ, UTD ratio, and subset minimization, are orthogonal methods to each other, we experiment with SPQR-REDQ for UTD ratio 1 and 20. A high UTD ratio can be interpreted as frequent gradient updates, thereby resulting in better performance. For each algorithm, we plot the average return as a solid curve and the standard deviation as a shaded area. The implementation of SPQR yields notable improvements in the performance of SAC-Ens and REDQ across all tasks, as well as under both UTD ratio 1 and 20 conditions. Specifically, SPQR-SAC-Ens performs **2** times better than SAC-Ens on the Hopper task. Also, SPQR-REDQ-UTD=1 and SPQR-REDQ-UTD=20 show **1.1** and **1.3** times better than REDQ-UTD=1 and REDQ-UTD=20 for the Walker2d and Ant tasks.

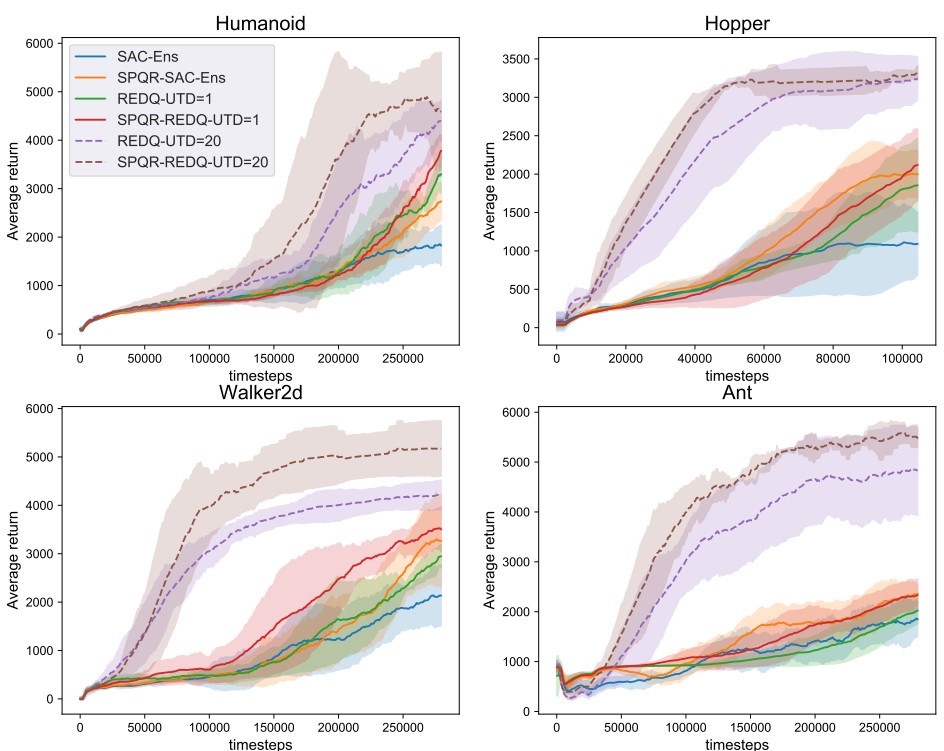

Figure 7: Average returns of SPQR-SAC-Ens, SPQR-REDQ-UTD=1, and SPQR-REDQ-UTD=20 on the MuJoCo environment. Averaged over 4 random seeds.

To investigate why SPQR performs better, we evaluate the bias of the Q-value computed by the Monte-Carlo algorithm. The average of the Q-bias decreases with SPQR, which can be interpreted as the relationship between the distribution of the ensemble becoming more independent. We also evaluate the ablation study for the UTD ratio. SPQR-SAC-Ens shows performance improvement and bias alleviation with low standard deviation as the UTD ratio increases although high UTD SAC causes large and unstable bias. In terms of diversity, we also evaluate the standard deviation with different $\beta$ for SPQR. As $\beta$ increases, the standard deviation for explored state-action also increases in all training timesteps. In summary, Q-ensemble independence is the key factor in improving the performance of SPQR. Further details and experimental results can be found in Appendix E.

**Offline Reinforcement Learning.** SAC-Min is formally constructed and evaluated in [2]. We follow the reported value and evaluation protocol from the paper. We consider baseline *behavior cloning* (BC), CQL, EDAC, and SAC-Min. The evaluation results of SPQR-SAC-Min compared with SAC-Min and other baselines are shown in Table 2. For most tasks, SPQR outperforms all baselines well. Specifically, for *-random dataset, SPQR improves SAC-Min by about **16%** when CQL and EDAC fail for *-random dataset. It implies that SPQR is robust for the quality of data. Since SPQR gives more conservatism, which shows identical behavior with increasing ensemble size $N$, SPQR-SAC-Min needs a smaller number of ensemble networks to outperform SAC-Min. For the hopper-medium, SAC-Min uses 500 Q-networks and SPQR uses only 50 Q-networks, which is about **90%** efficient in terms of computational cost. It implies that the independence regularization of the Q-ensemble becomes computationally efficient.

We also implement SPQR to CQL($\mathcal{H}$), denoted as SPQR-CQL-Min. Referring to the implementation details in the paper and the original baseline code of CQL, the original CQL uses a twin Q-function trick, which is equivalent to SAC-Min with $N = 2$. We modified CQL as CQL-Min to use $N$

Table 2: Normalized average returns of SPQR-SAC-Min and SPQR-CQL-Min on D4RL Gym, Franka Kitchen, and Antmaze tasks, averaged over 4 random seeds. Except for SPQR-*, the evaluated values of baselines are reported refer to [2, 21], and reproduced for *-full-replay tasks of CQL-Min. Reproduced values are indicated as *italic* font.

| Gym | BC | CQL-Min | EDAC | SAC-Min | SPQR-SAC-Min |
|---|---|---|---|---|---|
| walker2d-random | $1.3 \pm 0.1$ | 7.0 | $16.6 \pm 7.0$ | $21.7 \pm 0.0$ | $\mathbf{24.6 \pm 1.1}$ |
| walker2d-medium | $70.9 \pm 11.0$ | 74.5 | $92.5 \pm 0.8$ | $87.9 \pm 0.2$ | $\mathbf{98.4 \pm 2.0}$ |
| walker2d-expert | $108.7 \pm 0.2$ | $\mathbf{121.6}$ | $115.1 \pm 1.9$ | $116.7 \pm 0.4$ | $115.2 \pm 1.3$ |
| walker2d-medium-expert | $90.1 \pm 13.2$ | 98.7 | $114.7 \pm 0.9$ | $116.7 \pm 0.4$ | $\mathbf{118.2 \pm 0.7}$ |
| walker2d-medium-replay | $20.3 \pm 9.8$ | 32.6 | $87.1 \pm 2.3$ | $78.7 \pm 0.7$ | $\mathbf{87.8 \pm 2.5}$ |
| walker2d-full-replay | $68.8 \pm 17.7$ | *98.96* | $99.8 \pm 0.7$ | $94.6 \pm 0.5$ | $\mathbf{101.1 \pm 0.4}$ |
| halfcheetah-random | $2.2 \pm 0.0$ | $\mathbf{35.4}$ | $28.4 \pm 1.0$ | $28.0 \pm 0.9$ | $33.5 \pm 2.5$ |
| halfcheetah-medium | $43.2 \pm 0.6$ | 44.4 | $65.9 \pm 0.6$ | $67.5 \pm 1.2$ | $\mathbf{74.8 \pm 1.3}$ |
| halfcheetah-expert | $91.8 \pm 1.5$ | 104.8 | $106.8 \pm 3.4$ | $105.2 \pm 2.6$ | $\mathbf{112.8 \pm 0.3}$ |
| halfcheetah-medium-expert | $44.0 \pm 1.6$ | 62.4 | $106.3 \pm 1.9$ | $107.1 \pm 2.0$ | $\mathbf{114.0 \pm 1.6}$ |
| halfcheetah-medium-replay | $37.6 \pm 2.1$ | 46.2 | $61.3 \pm 1.9$ | $63.9 \pm 0.8$ | $\mathbf{69.7 \pm 1.4}$ |
| halfcheetah-full-replay | $62.9 \pm 0.8$ | *82.07* | $84.6 \pm 0.9$ | $84.5 \pm 1.2$ | $\mathbf{88.5 \pm 0.5}$ |
| hopper-random | $3.7 \pm 0.6$ | 7.0 | $25.3 \pm 10.4$ | $31.3 \pm 0.0$ | $\mathbf{35.6 \pm 1.4}$ |
| hopper-medium | $54.1 \pm 3.8$ | 86.6 | $\mathbf{101.6 \pm 0.6}$ | $100.3 \pm 0.3$ | $100.2 \pm 1.3$ |
| hopper-expert | $107.7 \pm 9.7$ | 109.9 | $110.1 \pm 0.1$ | $110.3 \pm 0.3$ | $\mathbf{112.0 \pm 0.2}$ |
| hopper-medium-expert | $53.9 \pm 4.7$ | 111.0 | $110.7 \pm 0.1$ | $110.1 \pm 0.3$ | $\mathbf{112.5 \pm 0.3}$ |
| hopper-medium-replay | $16.6 \pm 4.8$ | 48.6 | $101.0 \pm 0.5$ | $101.8 \pm 0.5$ | $\mathbf{104.9 \pm 0.7}$ |
| hopper-full-replay | $19.9 \pm 12.9$ | *104.85* | $105.4 \pm 0.7$ | $102.9 \pm 0.3$ | $\mathbf{109.1 \pm 0.4}$ |
| Average | 49.9 | *70.9* | 85.2 | 84.5 | $\mathbf{89.6}$ |
| **Franka Kitchen** | **BC** | **SAC-Min** | **BEAR** | **CQL-Min** | **SPQR-CQL-Min** |
| kitchen-complete | 33.8 | 15.0 | 0.0 | 43.8 | $\mathbf{47.9 \pm 12.3}$ |
| kitchen-partial | 33.8 | 0.0 | 13.1 | 49.8 | $\mathbf{54.2 \pm 7.2}$ |
| kitchen-mixed | 47.5 | 2.5 | 47.2 | 51.0 | $\mathbf{55.6 \pm 7.9}$ |
| Average | 38.4 | 5.8 | 20.1 | 48.2 | $\mathbf{52.6}$ |
| **Antmaze** | **BC** | **SAC-Min** | **BEAR** | **CQL-Min** | **SPQR-CQL-Min** |
| umaze | 65.0 | 0.0 | 73.0 | 74.0 | $\mathbf{93.3 \pm 4.7}$ |
| umaze-diverse | 55.0 | 0.0 | 61.0 | $\mathbf{84.0}$ | $80.0 \pm 0.0$ |
| medium-play | 0.0 | 0.0 | 0.0 | 61.2 | $\mathbf{80.0 \pm 8.2}$ |
| Average | 40.0 | 0.0 | 44.7 | 73.1 | $\mathbf{84.4}$ |

Q-networks to compute the Bellman target. Hence, the original CQL can be called CQL-Min with $N = 2$. We evaluate SPQR-CQL-Min on D4RL Franka Kitchen and Antmaze, one of the most complex and sparse tasks in D4RL.

In Table 2, we evaluate SPQR-CQL-Min with other baseline methods. SPQR-CQL-Min outperforms all baseline algorithms, notably 2.6 times better than BEAR. For Franka Kitchen tasks, SPQR **consistently** improves CQL by **10%** for all three tasks, which can be interpreted as robustness over the quality of data. Surprisingly, SPQR achieves the **maximum score** on 3 different Antmaze tasks, which outperforms all existing algorithms. Further details and results are presented in Appendix D and E.

**Can other diversification methods benefit from the SPQR?**  Since SPQR regularizes Q-ensemble independence, which is the orthogonal method with other diversification methods, we implement SPQR in EDAC to benefit from the combination of two orthogonal methods. We evaluate SPQR-EDAC for the tasks where EDAC outperforms the baseline algorithm SAC-Min. SPQR and SPQR-EDAC show large performance gains over SAC-Min, as shown in Table 3.

Table 3: Normalized average returns of SPQR-SAC-Min and SPQR-EDAC on D4RL Gym tasks, averaged over 4 random seeds. Except for SPQR-$*$, the evaluated values of baselines are reported. Refer to [2].

| Gym | EDAC | SAC-Min | SPQR-SAC-Min | SPQR-EDAC |
|---|---|---|---|---|
| walker2d-medium | $92.5 \pm 0.8$ | $87.9 \pm 0.2$ | $\mathbf{98.4 \pm 2.0}$ | $94.8 \pm 1.0$ |
| walker2d-medium-replay | $87.1 \pm 2.3$ | $78.7 \pm 0.7$ | $87.8 \pm 2.5$ | $\mathbf{89.6 \pm 1.8}$ |
| walker2d-full-replay | $99.8 \pm 0.7$ | $94.6 \pm 0.5$ | $101.1 \pm 0.4$ | $\mathbf{102.2 \pm 0.2}$ |
| halfcheetah-expert | $106.8 \pm 3.4$ | $105.2 \pm 2.6$ | $\mathbf{112.8 \pm 0.3}$ | $110.3 \pm 0.2$ |
| halfcheetah-full-replay | $84.6 \pm 0.9$ | $84.5 \pm 1.2$ | $\mathbf{88.5 \pm 0.5}$ | $86.8 \pm 0.2$ |
| hopper-medium | $101.6 \pm 0.6$ | $100.3 \pm 0.3$ | $100.2 \pm 1.3$ | $\mathbf{103.8 \pm 0.1}$ |
| hopper-medium-expert | $110.7 \pm 0.1$ | $110.1 \pm 0.3$ | $\mathbf{112.5 \pm 0.3}$ | $111.7 \pm 0.0$ |
| hopper-full-replay | $105.4 \pm 0.7$ | $102.9 \pm 0.3$ | $\mathbf{109.1 \pm 0.4}$ | $108.2 \pm 0.5$ |

# 6  Conclusions

We propose SPQR, a Q-ensemble independence regularization framework for ensemble Q-learning by applying random matrix theory and a spiked random model. We theoretically propose a tractable loss for Q-ensemble independence. The proposed theoretical approach does not require any prior assumption on the Q-function distribution, such as uniform distribution or i.i.d. In addition, we present the difference between diversity and independence and empirically show the orthogonality of the diversification method and SPQR. Furthermore, SPQR serves as a conservatism control framework for ensemble Q-learning by effectively regularizing independence with various empirical evidence. We also empirically show that SPQR is computationally efficient and outperforms many existing ensemble Q-learning algorithms on various online and offline RL benchmark tasks.

# Acknowledgements

We appreciate Minyoung Kim for creating an illustration for our paper, as well as Jaehoon Choi and Yejin Kim for writing advice. We also thank the anonymous reviewers for their insightful feedback and reviews. This work is in part supported by National Research Foundation of Korea (NRF, 2021R1A2C2014504(20%), 2021M3F3A2A02037893(20%)), Institute of Information & communications Technology Planning & Evaluation (IITP, 2021-0-00106(20%), 2021-0-01059(20%), 2021-0-02068(20%)) grant funded by the Ministry of Science and ICT (MSIT), INMAC and BK21 FOUR Program.

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

# A  Review for Detection Theory and Spiked Random Model

We call *strong detection* when we perfectly detect the presence of a spike signal and *weak detection* when we detect the presence of a spike signal with some error by hypothesis testing. A formal definition for a strong detection can be constructed under the notion of *contiguity* by [23]. The contiguity of the spiked Wishart model is proven by the second moment method, which means that we cannot distinguish without error whether data is pure noise or contains some signal.

**Definition A.1** (Contiguity). Let $P_n$ and $Q_n$ be two sequences of probability measures defined on the sequence of measurable spaces $(\Omega_n, \mathcal{F}_n)$. $Q_n$ is *contiguous* with respect to the sequence $P_n$ if $P_n(A_n) \to 0$ implies $Q_n(A_n) \to 0$ as $n \to \infty$ for every sequence of measurable sets $A_n \in \mathcal{F}_n$, denoted as $Q_n \lhd P_n$. The sequences $P_n$ and $Q_n$ are mutually contiguous if both $Q_n \lhd P_n$ and $P_n \lhd Q_n$ holds, denoted as $Q_n \lhd \rhd P_n$

**Lemma A.2** (Le Cam's First Lemma). *Let $P_n$ and $Q_n$ be two sequences of probability measures defined on the sequence of measurable spaces $(\Omega_n, \mathcal{F}_n)$. Then the following statements are equivalent:*
*1. $Q_n \lhd P_n$*

*2. If $dP_n/dQ_n \xrightarrow{Q_n} U$ along a subsequence, then $P(U > 0) = 1$*

*3. If $dQ_n/dP_n \xrightarrow{P_n} V$ along a subsequence, then $\mathbb{E}[V] = 1$*

*4. For any statistics $T_n : \Omega_n \mapsto \mathbb{R}^k$, If $T_n \xrightarrow{P_n} 0$, then $T_n \xrightarrow{Q_n} 0$*

**Lemma A.3** (Second moment method). *Let $P_n$ and $Q_n$ be two sequences of probability measures defined on the sequence of measurable spaces $(\Omega_n, \mathcal{F}_n)$, and Radon-Nikodym derivative of $P_n$ with respect to $Q_n$ as $L_n \equiv dP_n/dQ_n$. If $\limsup \mathbb{E}_{Q_n}[L_n^2] < \infty$, then $P_n \lhd Q_n$*

**Lemma A.4.** *For spiked wishart model following definition 3.2, $K_u^4 K_v^4 \psi^2 < 1$, the families of distributions $\mathbb{P}_0$ and $\mathbb{P}_\psi$, indexed by $M, N$, are mutually contiguous in the limit $N \to \infty, M/N \to \omega$*

**Hypothesis Testing for Weak Detection.**  To distinguish whether spiked Wishart model $Y$ follows $\mathbb{P}_\psi$ or $\mathbb{P}_0$, detection error is inevitable since contiguity holds. Even if the perfect test is not available, it needs to be better than a random guess, which is achievable by constructing a hypothesis test. Let $(\Omega, \mathcal{F}, \hat{\rho})$ be a probability space, $(\mathcal{X}, \mathcal{B})$ be a topological space with Borel $\sigma$-algebra $\mathcal{B}$, and $\Theta$ to be a set of parameters. For observed random variable $X : \Omega \mapsto \mathcal{X}$ with $X \sim P_\theta, \theta \in \Theta$, and fixed $\hat{\theta} \neq 0$, Objective of hypothesis testing is to distinguish between two hypotheses, called *null hypothesis $H_0$* and *alternative hypothesis $H_1$*

$$H_0 : \theta = 0, H_1 : \theta = \hat{\theta} \tag{3}$$

Constructing a measurable function $T : \mathcal{X} \mapsto \{0, 1\}$ to represent test, $T(X) = 0$ means accept $H_0$ and $T(X) = 1$ means accept $H_1$. The error of test $T$ is the sum of Type-I error, false positive, and Type-II error, false negative, stated as,

$$err(T) = P_{\hat{\theta}}(T(X) = 0) + P_0(T(X) = 1) \tag{4}$$

Optimal test $T^*$ is defined as minimizing Type-II error subject to the constraint that Type-I error is no greater than a predetermined level $\alpha$. $T^*$ must have a form of likelihood ratio test by following *Neyman-Pearson Lemma* and propositions from [19].

**Lemma A.5** (Neyman-Pearson Lemma). *Let null hypothesis $H_0$ with probability measure $p_0$ and density $f_0$, alternative hypothesis $H_1$ with probability measure $p_1$ and density $f_1$, density ratio $r(x) = f_1(x)/f_0(x)$, and constant $k \in \mathbb{R}$ such that $p_0\{r(x) > k\} \leq \alpha$ for a given significance level $\alpha$. There exist an optimal test function $T^* : \mathbb{R}^D \to \mathbb{R}$, with a mild continuity assumption, such that*

$$T^*(X) = \mathbb{1}(f_1(x)/f_0(x) > k) = \mathbb{1}(r(x) > k)$$

To construct an optimal test by applying Neyman-Pearson Lemma, we define a likelihood ratio, *Radon-Nikodym derivative* between $\mathbb{P}_\psi$ and $\mathbb{P}_0$ as $L(Y; \psi)$ by the following equations. For convience, let *Hamiltonian $H : \mathbb{R}^{N+M} \to \mathbb{R}$* be as follows:

$$-H(u, v) = \sum_{i,j} \sqrt{\frac{\psi}{N}} Y_{ij} u_i v_j - \frac{\psi}{2N} u_i^2 v_j^2 \tag{5}$$

$$dP_u^{\otimes N}(u) = dP_u(u_1) \otimes \cdots \otimes dP_u(u_N) \tag{6}$$

where $\otimes$ represents a tensor product, $L(Y; \psi)$ has form of

$$L(Y; \psi) = \frac{d\mathbb{P}_\psi}{d\mathbb{P}_0} = \int e^{-H(u,v)} dP_u^{\otimes N}(u) dP_v^{\otimes M}(v) \tag{7}$$

The optimal test for a spiked Wishart model is given as follows:

**Lemma A.6.** *Let us assume that a spiked Wishart model is defined in definition 3.2. Let null hypothesis $H_0 : Y \sim \mathbb{P}_0$, alternative hypothesis $H_1 : Y \sim \mathbb{P}_\psi$, likelihood ratio $L$ between $\mathbb{P}_\psi$ and $\mathbb{P}_0$, and define tests $T : \mathbb{R}^{N \times M} \to \{0, 1\}$. By Lemma A.5, optimal test is,*

$$T^*(Y) = \mathbb{1}(\log L > 0) = \begin{cases} H_0 : Y \sim \mathbb{P}_0, & \text{if } \log L \le 0 \\ H_1 : Y \sim \mathbb{P}_\psi, & \text{if } \log L > 0 \end{cases}$$

Proof of A.6 is given in Section B.

**Corollary A.7.** *For optimal test $T^*$, Error of $T^*$ is*

$$err_{M,N}^*(T^*) = P_0(\log L(Y; \psi) > 0) + P_\psi \log L(Y; \psi) \le 0) = 1 - D_{TV}(\mathbb{P}_\psi, \mathbb{P}_0)$$

*For $\psi \ge 0$ such that $K_u^4 K_v^4 \psi^2 < 1$*

$$\lim_{N \to \infty} err_{M,N}^*(T^*) = 1 - \lim_{N \to \infty} D_{TV}(\mathbb{P}_\psi, \mathbb{P}_0) = erfc\left(\frac{1}{4}\sqrt{-\log(1 - \psi^2)}\right)$$

*where $erfc(x) = \frac{2}{\sqrt{\pi}} \int_x^\infty e^{-t^2} dt$*

The error for optimal test $T^*$ is stated at Corollary A.7. Likelihood ratio $\log L$ is used for the optimal test to distinguish whether data $Y$ in ensembles are sampled from i.i.d prior $\mathbb{P}_0$. However, $\log L$ is intractable since information about $\psi$ and $u, v$ is not given. Therefore, we need an alternative criterion for optimal test $T^*$ that does not require prior knowledge to make it tractable with ensemble Q-learning. Intuitively, although we cannot know the exact form of $\mathbb{P}_\psi$ and $\mathbb{P}_0$, its eigenvalue distribution can be described by ESD and Wigner's semicircle law. The expectation of $\log L$ becomes KL divergence between $\mathbb{P}_\psi$ and $\mathbb{P}_0$, followed by the definition of $\log L$. By integrating these intuitions, intractable criterion $\log L$ can be restated as KL divergence between ESD and Wigner's semicircle law, which we prove in Section B.

*Proof of Lemma A.2.* The proof is exactly same as the Lemma 6.4 of [33] ∎

*Proof of Lemma A.3.* For any event $A \in \mathcal{F}_n$, by Cauchy-Schwarz inequality,

$$P_n(A) = \int \mathbb{1}_A(w) dP_n(w) = \int \mathbb{1}_A(w) L_n(w) dQ_n(w)$$

$$\le \sqrt{\int L_n^2 dQ_n} \cdot \sqrt{\int \mathbb{1}_A dQ_n} = \sqrt{\mathbb{E}_{Q_n}[L_n^2]} \cdot \sqrt{Q_n(A)}$$

$\therefore$ If $\limsup \mathbb{E}_{Q_n}[L_n^2] < \infty$, then $Q_n(A_n) \to 0$ implies $P_n(A_n) \to 0$, concludes to $P_n \triangleleft Q_n$ ∎

*Proof of Lemma A.4.* The proof is exactly same as the Theorem 4 of [4] ∎

# B Proofs

## B.1 Technical Lemmas and Proof

Before proving our theoretical results, we present some lemmas to clear the description.

**Lemma B.1** (Convergence). *Let $\psi \geq 0$ such that $K_u^4 K_v^4 \psi^2 < 1$, then in the limit $N \to \infty$, where $\leadsto$ denotes convergence in distribution,*

$$\log L(Y; \psi) \leadsto \mathcal{N}\left(\pm \frac{1}{4}\log(1 - \psi^2), -\frac{1}{2}\log(1 - \psi^2)\right) \tag{8}$$

*The sign $+$ means $Y \sim \mathbb{P}_0$, $-$ means $Y \sim \mathbb{P}_\psi$*

**Lemma B.2** (Concentration). *for fixed $u^*, v^*$, $Y = \sqrt{\frac{\psi}{N}} u^* v^{*T} + W$, for every $t \geq 0$,*

$$Pr(|\log L(Y; \psi) - \mathbb{E}[\log L(Y; \psi)]| \geq Nt) \leq 2\exp(-Nt^2/2\psi K_u^2 K_v^2)$$

**Lemma B.3** (Weyl's Lemma). *If $X$ is rotational invariance, $P[X] = g(Tr(X), Tr(X^2), \ldots, Tr(X^N))$*

**Lemma B.4** (Vandermonde determinant). *For GOE $Y = O\Lambda O^T$, the relationship between joint probability density function of matrix entity and eigenvalues is given as,*

$$P(Y_{11}, Y_{12}, \cdots, Y_{NN})dY_{11}dY_{12}\cdots dY_{NN} = P(\lambda_1, \lambda_2, \cdots, \lambda_N, \{O\})d\lambda_1 d\lambda_2 \cdots d\lambda_N d\{O\}$$
$$P(\lambda_1, \lambda_2, \cdots, \lambda_N, \{O\}) = P(Y_{11}, Y_{12}, \cdots, Y_{NN})|J(Y \to \lambda, \{O\})|$$
$$\text{where } J(Y \to \lambda, \{O\}) = dY_{11}dY_{12}\cdots dY_{NN}/d\lambda_1 d\lambda_2 \cdots d\lambda_N d\{O\}$$

*Then vandermonde determinant become*

$$|J(Y \to \lambda, \{O\})| = |J(Y \to \lambda)| = |dY/d\lambda| = \prod_{i<j}|\lambda_i - \lambda_j|$$

*Proof of Lemma B.1.* The proof is exactly same as the Theorem 3.1 of [11] ∎

*Proof of Lemma B.2.* The proof is exactly same as the Lemma 3.14 of [11] ∎

*Proof of Lemma B.3.* The proof is exactly same as the Lemma 1.1 of [9] ∎

*Proof of Lemma B.4.* The proof is exactly same as the procedure in Chapter 3 of [26] ∎

*Proof of lemma A.6.* Using Lemma B.1 and Lemma A.5, $\log L$ can distinguish at $\log L = 0$ as $N \to \infty$ because of symmetry of Gaussian mean, i.e $k = 1$ for Neyman-Pearson Leamma. Therefore, optimal test $T^* = \mathbb{1}(\log L > 0)$ ∎

## B.2 Proof of Main Theorem

**Theorem 4.1.** *Following definition 3.2, as $N \to \infty$ with probability at least $1 - \delta$, following test $T(\lambda)$ is optimal.*

$$T(\lambda) = \mathbb{1}(D_{KL}(p_\psi(\lambda), p_{sc}(\lambda)) \geq \varepsilon) = \begin{cases} H_0 : Y \sim \mathbb{P}_0 \\ H_1 : Y \sim \mathbb{P}_\psi \end{cases}$$

*Proof.* By Lemma B.3 and Lemma B.4, For spiked wishart model $Y \sim \mathbb{P}_\psi$

$$p_\psi(\lambda) = P(\lambda_1, \lambda_2, \cdots, \lambda_N) = \int dO P(\lambda_1, \lambda_2, \cdots, \lambda_N, \{O\})$$

$$= \int dO P(Y_{11}, Y_{12}, \cdots, Y_{NN})|J(Y \to \lambda, \{O\})|$$

$$= |J_\psi|\mathbb{P}_\psi(Y)\int dO, \text{ where } \int dO = \text{constant(volume)} > 0$$

If $D_{KL}(p_\psi(\lambda), p_{sc}(\lambda)) < \varepsilon$,

$$D_{KL}(\mathbb{P}_\psi, \mathbb{P}_0) = |J_\psi| D_{KL}(p_\psi(\lambda), p_{sc}(\lambda)) + |J_\psi| \log \frac{C|J_\psi|}{|J_0|}, C > 0$$

$$\Rightarrow D_{KL}(\mathbb{P}_\psi, \mathbb{P}_0) < |J_\psi|\varepsilon + |J_\psi| \log \frac{C|J_\psi|}{|J_0|} = \tilde{\varepsilon}, \quad \because |J_\psi| > 0$$

Let $\Delta = \log L - \mathbb{E}[\log L]$, $\hat{t} = Nt$. By Lemma B.2,

$$Pr(|\Delta| \geq \hat{t}) \leq 2\exp(-\hat{t}^2/2\psi N K_u^2 K_v^2)$$

Let $\delta = 2\exp(-\hat{t}^2/2\psi N K_u^2 K_v^2)$. Then, with probability at least $1 - \delta$,

$$\Delta \in [-\Delta_{max}, \Delta_{max}]$$
$$\Delta_{max} = \sqrt{2N\psi K_u^2 K_v^2 \log(2/\delta)}$$

Since $D_{KL}(\mathbb{P}_\psi, \mathbb{P}_0) < \tilde{\varepsilon}$, then,

$$\log L = \mathbb{E}[\log L] + \Delta = D_{KL}(\mathbb{P}_\psi, \mathbb{P}_0) + \Delta < \tilde{\varepsilon} + \Delta$$

(9)

Choose $\tilde{\varepsilon} = -\Delta - \delta'$, $\delta' > 0$. Then,

$$\log L < -\delta' < 0$$

$\therefore$ with probability at least $1 - \delta$, test $T(\lambda)$ is optimal test $T^*$ as $N \to \infty$ ∎

# C Algorithm Details

## C.1 Detailed Algorithm for SPQR

We describe a detailed version of SPQR for Algorithm 3. A detailed description for computing SPQR loss is shown in Algorithm 4. Table 4 shows functions used in implementing baseline ensemble Q-learning. For Figure 8, we visualize an illustrative example of SPQR.

---

**Algorithm 3** SPQR

---

**Input:** discount $\gamma \in [0, 1)$, temperature parameter $\alpha$, ensemble size $N$, SPQR loss gain $\beta$, polyak $\tau$, batch size $|B|$

Initialize policy parameter $\theta$, Q-function parameters $\{\phi_i\}_{i=1}^N$, target Q-function parameters $\{\phi_i' \leftarrow \phi_i\}_{i=1}^N$, online/offline replay buffer $\mathcal{D}$

**repeat**
    **if** $\mathcal{D}$ not offline dataset **then**
        Take action from $a_t \sim \pi_\theta(\cdot|s_t)$, observe reward $r_t$, next state $s_{t+1}$
        $\mathcal{D} \leftarrow \mathcal{D} \cup \{(s_t, a_t, r_t, s_{t+1})\}$
    **end if**
    $B \leftarrow \{(s_j, a_j, r_j, s_j')\}_{j=1}^{|B|}$ from $\mathcal{D}$         ▷ Sample a mini-batch
    $y(r, s') \leftarrow r + \gamma \left( Ens_{tar}(\{Q_{\phi_i'}(s', a')\}_{i=1}^N) - \alpha \log \pi_\theta(a'|s') \right), a' \sim \pi_\theta(\cdot|s')$ ▷ Compute Q-target
    $\mathcal{L}_{SPQR} \leftarrow \mathcal{L}_{SPQR}(\{\phi_i\}_{i=1}^N)$         ▷ Compute SPQR loss using Algorithm 4
    **for** $i = 1, \ldots, N$ **do**
        $\mathcal{L}_{\phi_i} \leftarrow \frac{1}{|B|}\sum_{(s,a,r,s')}(Q_{\phi_i}(s, a) - y(r, s'))^2 + \beta\mathcal{L}_{SPQR}$         ▷ Compute Q loss
        Update $\phi_i$ with gradient descent using $\nabla_{\phi_i}\mathcal{L}_{\phi_i}$
        Update $\phi_i' \leftarrow \tau\phi_i' + (1 - \tau)\phi_i$
    **end for**
    $\mathcal{L}_\theta \leftarrow \frac{1}{|B|}\sum_{s\in B}\left( Ens_{eval}(\{Q_{\phi_i}(s, \tilde{a})\}_{i=1}^N) - \alpha \log \pi_\theta(\tilde{a}|s) \right), \tilde{a} \sim \pi_\theta(\cdot|s)$     ▷ Compute policy loss
    Update $\theta$ with gradient ascent using $\nabla_\theta\mathcal{L}_\theta$
**until**
**Output:** parameter $\theta$, $\{\phi_i\}_{i=1}^N$, $\{\phi_i'\}_{i=1}^N$

---

Table 4: Target and evaluation functions used in Ensemble Q-learning algorithms

| Algorithm | $Ens_{tar}(\{Q_{\phi_i'}(s', a')\}_{i=1}^N)$ | $Ens_{eval}(\{Q_{\phi_i}(s, \tilde{a})\}_{i=1}^N)$ |
|---|---|---|
| SAC-Ens | $\frac{1}{N}\sum_{i=1}^N Q_{\phi_i'}(s', a')$ | $\frac{1}{N}\sum_{i=1}^N Q_{\phi_i}(s, \tilde{a})$ |
| REDQ | $\min_{i\in\mathcal{M}} Q_{\phi_i'}(s', a'), \mathcal{M} \subset \{1, \ldots, N\}$ | $\frac{1}{N}\sum_{i=1}^N Q_{\phi_i}(s, \tilde{a})$ |
| SAC-Min | $\min_{i=1,\ldots,N} Q_{\phi_i'}(s', a')$ | $\min_{i=1,\ldots,N} Q_{\phi_i}(s, \tilde{a})$ |
| CQL-Min | $\min_{i=1,\ldots,N} Q_{\phi_i'}(s', a')$ | $\min_{i=1,\ldots,N} Q_{\phi_i}(s, \tilde{a})$ |

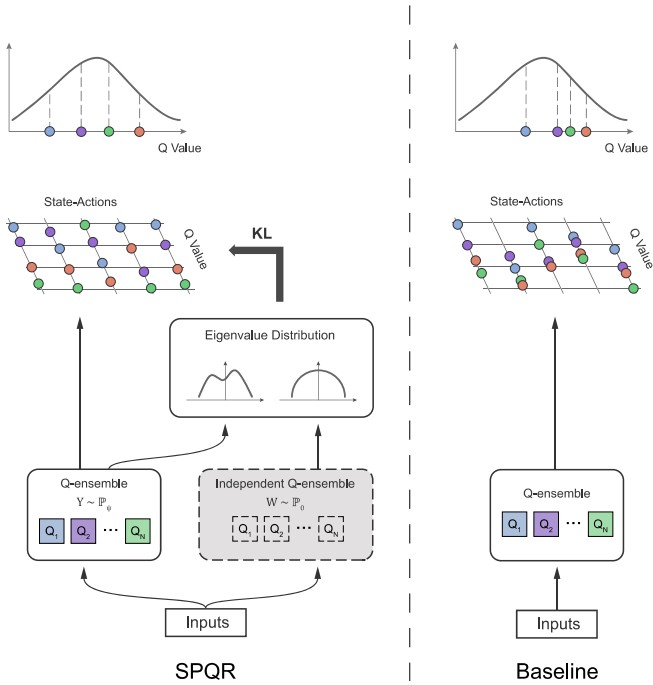

Figure 8: Illustrative example of SPQR compared to baseline methods.

## C.2 Detailed Algorithm for SPQR Loss

Algorithm 4 gives a detailed explanation for computing SPQR loss. We build a symmetric random matrix with a randomly-ordered Q-value. Our goal is to generate a symmetric Q-value matrix filled with an ensemble size of N. To avoid training or capturing the order of a matrix, we shuffle the filling order. This can be achieved by sorting the list of Q-networks $l_k$. In order to create a symmetric matrix of maximum size, the upper triangle part and the corresponding lower triangle part are equally filled. The size of the matrix is set as $D = \lfloor \frac{\sqrt{1+8N}-1}{2} \rfloor$. D can be found by finding the largest integer that satisfies $\frac{D^2+D}{2} \leq N$. To avoid numerical instability when computing KL divergence, we formulate Wigner's semicircle distribution as like $\varepsilon$-soft, in which all probability distribution value is at least $\varepsilon$, and we call it a **soft** Wigner's semicircle distribution. $\rho$ gives weight between $\varepsilon$-soft distribution and Wigner's semicircle distribution. We fix $\rho = 0.5$ and $\varepsilon = 0.01$.

---

**Algorithm 4** SPQR Loss

---

**Input:** List of Q-network $l_k = \{Q_{\phi_i}(s', a')\}_{i=1}^N$

$Y \in \mathbb{R}^{D \times D}, D = \lfloor \frac{\sqrt{1+8N}-1}{2} \rfloor$:

$Y_{pq} \leftarrow \begin{cases} l_{Dp+q}, & \text{if } p \geq q \\ Y_{qp}, & \text{otherwise} \end{cases}$        ▷ Build random matrix

$Y \leftarrow (Y - \mu_Y)/\sigma_Y$        ▷ Normalizing

$\{\lambda_i\}_{i=1}^N \leftarrow Eigen(\frac{1}{\sqrt{N}}Y)$        ▷ Compute eigenvalue

$p_{esd}(\lambda) = \frac{1}{N} \sum_i \delta(\lambda - \lambda_i)$        ▷ Define ESD where $\delta$ is dirac delta function

$p_{wigner}(\lambda) = \begin{cases} \rho \frac{\sqrt{4-\lambda^2}}{2\pi}, & \text{if } \lambda \leq 2 \\ (1-\rho)\varepsilon, & \text{otherwise} \end{cases}$        ▷ Define soft Wigner's semicircle distribution

$\mathcal{L}_{SPQR} \leftarrow \frac{1}{|B|} \sum_j \sum_i p_{esd}(\lambda_i) \log \frac{p_{esd}(\lambda_i)}{p_{wigner}(\lambda_i)}$        ▷ Compute SPQR loss

**Output:** SPQR loss $\mathcal{L}_{SPQR}$

---

## C.3 Eigenvalue Backpropagation

To use $D_{KL}(p_\psi(\lambda), p_{sc}(\lambda))$ as a regularization loss, we need to backpropagate loss gradient through eigenvalue. In Theorem C.1, we construct a backpropagation equation about eigenvalue refer to [16] and [14]. For practical implementation, using `torch.eigvals` and `torch.eigvalsh` gives numerically stable eigenvalue backpropagation by solving $det(X - \lambda I) = 0$, not solving $Xu = \lambda u$ geometrically.

**Theorem C.1.** *Let $L : \mathbb{R}^d \to \mathbb{R}$ be a loss function, and $X = U\Lambda U^T$ with $X \in \mathbb{R}^{m \times m}$, such that $U^T U = I$ and $\Lambda$ possessing diagonal structure. Using the notation $A \circ B$ as a Hadamard product of two matrices, which $(A \circ B)_{ij} = A_{ij} B_{ij}$ and $A_{sym} = \frac{1}{2}(A^T + A), A_{diag} = I \circ A$ Then*

$$d\Lambda = (U^T dXU)_{diag}$$

$$dU = U(K^T \circ (U^T dXU)_{sym})$$

$$K_{ij} = \begin{cases} \dfrac{1}{\lambda_i - \lambda_j}, & i \neq j \\ 0, & i = j \end{cases}$$

*The resulting partial derivatives are*

$$\frac{\partial L}{\partial X} = U \left\{ \left( K^T \circ \left( U^T \frac{\partial L}{\partial U} \right)_{sym} \right) + \left( \frac{\partial L}{\partial \Lambda} \right)_{diag} \right\} U^T$$

*Proof.* Let $X = U\Lambda U^T, U^T U = I$, then $XU = U\Lambda$. Differentiation gives

$$dXU + XdU = dU\Lambda + Ud\Lambda$$

Define $dC = U^T dU$, so that $dU = UdC$, then

$$dXU + U\Lambda dC = UdC\Lambda + Ud\Lambda$$

Multiplying $U^T$ to both sides and rearranging gives

$$dC\Lambda - \Lambda dC + d\Lambda = U^T dXU$$

Let $E_{ij} = \lambda_j - \lambda_i$, then

$$dC\Lambda - \Lambda dC = E \circ dC$$

Rearranging gives,

$$E \circ dC + d\Lambda = U^T dXU$$

Since the diagonal of $E$ are zero,

$$d\Lambda = I \circ (U^T dXU) = (U^T dXU)_{diag}$$

So,

$$dC = K \circ (U^T dXU) \Rightarrow dU = U(K \circ (U^T dXU)), \text{ where } K_{ij} = \begin{cases} \dfrac{1}{\lambda_i - \lambda_j}, & i \neq j \\ 0, & i = j \end{cases}$$

$$\therefore d\Lambda = I \circ (U^T dXU) = (U^T dXU)_{diag}, \quad dU = U(K^T \circ (U^T dXU)_{sym})$$

$\blacksquare$

## D   Implementation Details

### D.1   Online RL

We implement SPQR to SAC-Ens and REDQ on the REDQ code, SPQR to SAC-Min on the SAC-Min code, and SPQR to CQL on the CQL code. For online MuJoCo environment experiments, we implement SPQR-SAC-Ens and SPQR-REDQ. We use $*$-v2 version of MuJoCo environments and train 12.5k timesteps for Hopper and 30k for Humanoid, Walker2d, and Ant. Hyperparameters shown in Table 5 are the same with REDQ implementation, including the size of a subset $M = 2$ for REDQ and SPQR-REDQ. For fair evaluation, we fix ensemble size $N$ for each corresponding baseline algorithm, $N = 20$ for SAC-Ens and SPQR-SAC-Ens, and $N = 10$ for REDQ and SPQR-REDQ. We use annealing beta with a linear decay to stabilize training.

Table 5: Hyperparameters used in the Mujoco environments

| Hyperparameter | value |
| --- | --- |
| discount factor $\gamma$ | 0.99 |
| initial temperature parameter $\alpha$ | 0.2 |
| Q learning rate | 3e-4 |
| policy learning rate | 3e-4 |
| hidden layer width | 256 |
| hidden layer depth | 2 |
| batch size $|B|$ | 256 |
| polyak parameter $\tau$ | 0.995 |
| optimizer | Adam |

Table 6: Algorithm hyperparameters used in the Mujoco environments

| Task | SPQR-SAC-Ens $(N, \beta)$ | SPQR-REDQ $(N, \beta)$ |
| --- | --- | --- |
| Hopper | 20,0.5 | 10,0.5 |
| Humanoid | 20,2.0 | 10,2.0 |
| Walker2d | 20,0.1 | 10,0.1 |
| Ant | 20,2.0 | 10,2.0 |

### D.2   Offline RL

SPQR-SAC-Min is implemented based on SAC-Min and evaluated on D4RL MuJoCo Gym tasks with three environments, walker2d, halfcheetah, and hopper, each having separated six datasets. Each dataset is built with a different behavior policy. *random* uses 1M samples generated from the randomly initialized policy. *expert* uses 1M samples generated from a policy trained to completion with SAC. *medium* uses 1M samples generated from a policy that performs approximately 1/3 of the expert policy. *medium-replay* uses the replay buffer of a policy that performs as much as the medium agent. *full-replay* uses 1M samples from the final replay buffer of the expert policy. *medium-expert* uses a 50-50 split of medium and expert data, slightly less than 2M samples total. For fair evaluation with SAC-Min and EDAC, $*$-v2 datasets are used for all tasks, and the same hyperparameter with SAC-Min and EDAC, as shown in Table 7.

We use normalized score by calculating $100 \times \dfrac{raw\_score - random\_score}{expert\_score - random\_score}$, proposed evaluation metric from [13]. Algorithmic hyperparameter $\beta$ is shown in Table 8. We fix $\beta$ over similar tasks to show the robustness of SPQR. Also, we use a smaller ensemble size $N$ for SPQR-SAC-Min rather than $N$ for SAC-Min. For the hopper-$*$ dataset, SPQR drastically reduces ensemble size $N$, using only 10% of $N$ for SAC-Min. For SPQR-EDAC, we use the same hyperparameter with SPQR-SAC-Min and $\eta = 0.5$ for EDAC.

SPQR-CQL-Min is implemented based on CQL($\mathcal{H}$), called CQL-Min, and evaluated on D4RL Franka Kitchen and Antmaze tasks, one of the most complicated tasks in the D4RL environment. Since there is no information about any hyperparameters for evaluating Franka Kitchen using CQL in the official

paper and code, we use hyperparameters given for evaluating Antmaze in the CQL paper, shown in Table 7. Like our efforts at SPQR-SAC-Min to test the SPQR's robustness, we fix all hyperparameters $\beta$, as shown in Table 8. For all D4RL tasks, we use annealing beta with an exponential decay to stabilize training.

Table 7: Hyperparameters used in the D4RL tasks

| Hyperparameter | Gym | Franka Kitchen | Antmaze |
|---|---|---|---|
| discount factor $\gamma$ | 0.99 | 0.99 | 0.99 |
| initial temperature parameter $\alpha$ | 1.0 | 1.0 | 1.0 |
| Q learning rate | 3e-4 | 3e-4 | 3e-4 |
| policy learning rate | 3e-4 | 1e-4 | 1e-4 |
| hidden layer width | 256 | 256 | 256 |
| hidden layer depth | 3 | 3 | 3 |
| batch size $|B|$ | 256 | 256 | 256 |
| polyak parameter $\tau$ | 0.995 | 0.995 | 0.995 |
| optimizer | Adam | Adam | Adam |
| lagrange threshold (for CQL) | - | 5.0 | 5.0 |

Table 8: Algorithmic hyperparameter used in the D4RL Gym and Franka Kitchen tasks. SAC-Min and CQL-Min follow reported values. Refer to [2] and [21]

| Gym | SAC-Min ($N$) | SPQR-SAC-Min ($N$) | SPQR-SAC-Min ($\beta$) |
|---|---|---|---|
| walker2d-random | 20 | 20 | |
| walker2d-medium | 20 | 10 | |
| walker2d-expert | 100 | 100 | |
| walker2d-medium-expert | 20 | 20 | |
| walker2d-medium-replay | 20 | 10 | |
| walker2d-full-replay | 20 | 20 | |
| halfcheetah-random | 10 | 3 | |
| halfcheetah-medium | 10 | 3 | |
| halfcheetah-expert | 10 | 10 | |
| halfcheetah-medium-expert | 10 | 10 | 0.1 |
| halfcheetah-medium-replay | 10 | 10 | |
| halfcheetah-full-replay | 10 | 3 | |
| hopper-random | 500 | 50 | |
| hopper-medium | 500 | 50 | |
| hopper-expert | 500 | 200 | |
| hopper-medium-expert | 200 | 50 | |
| hopper-medium-replay | 200 | 50 | |
| hopper-full-replay | 200 | 50 | |
| Franka Kitchen | CQL-Min ($N$) | SPQR-CQL-Min ($N$) | SPQR-CQL-Min ($\beta$) |
| kitchen-complete | 2 | 6 | |
| kitchen-partial | 2 | 6 | 2.0 |
| kitchen-mixed | 2 | 6 | |
| Antmaze | CQL-Min ($N$) | SPQR-CQL-Min ($N$) | SPQR-CQL-Min ($\beta$) |
| umaze | 2 | 6 | |
| umaze-diverse | 2 | 6 | 2.0 |
| medium-play | 2 | 3 | |
| medium-diverse | 2 | 3 | |

# E Further Experimental Results

## E.1 Evaluation Result with More Baselines

Table 9: Normalized average returns of SPQR-SAC-Min, SPQR-EDAC and SPQR-CQL-Min on D4RL Gym, Franka Kitchen, and Antmaze tasks, averaged over 4 random seeds. Except for SPQR-∗, the evaluated values of baselines are reported. Refer to [2],[20], and [21], and reproduced. Reproduced values are indicated as *italic* font.

| Gym | BC | CQL-Min | EDAC | SAC-Min | SPQR-SAC-Min | SPQR-EDAC |
|---|---|---|---|---|---|---|
| walker2d-random | $1.3 \pm 0.1$ | 7.0 | $16.6 \pm 7.0$ | $21.7 \pm 0.0$ | $\mathbf{24.6 \pm 1.1}$ | - |
| walker2d-medium | $70.9 \pm 11.0$ | 74.5 | $92.5 \pm 0.8$ | $87.9 \pm 0.2$ | $\mathbf{98.4 \pm 2.0}$ | $94.8 \pm 1.0$ |
| walker2d-expert | $108.7 \pm 0.2$ | $\mathbf{121.6}$ | $115.1 \pm 1.9$ | $116.7 \pm 0.4$ | $115.2 \pm 1.3$ | - |
| walker2d-medium-expert | $90.1 \pm 13.2$ | 98.7 | $114.7 \pm 0.9$ | $116.7 \pm 0.4$ | $\mathbf{118.2 \pm 0.7}$ | - |
| walker2d-medium-replay | $20.3 \pm 9.8$ | 32.6 | $87.1 \pm 2.3$ | $78.7 \pm 0.7$ | $87.8 \pm 2.5$ | $\mathbf{89.6 \pm 1.8}$ |
| walker2d-full-replay | $68.8 \pm 17.7$ | *98.96* | $99.8 \pm 0.7$ | $94.6 \pm 0.5$ | $101.1 \pm 0.4$ | $\mathbf{102.2 \pm 0.2}$ |
| halfcheetah-random | $2.2 \pm 0.0$ | $\mathbf{35.4}$ | $28.4 \pm 1.0$ | $28.0 \pm 0.9$ | $33.5 \pm 2.5$ | - |
| halfcheetah-medium | $43.2 \pm 0.6$ | 44.4 | $65.9 \pm 0.6$ | $67.5 \pm 1.2$ | $\mathbf{74.8 \pm 1.3}$ | - |
| halfcheetah-expert | $91.8 \pm 1.5$ | 104.8 | $106.8 \pm 3.4$ | $105.2 \pm 2.6$ | $\mathbf{112.8 \pm 0.3}$ | $110.3 \pm 0.2$ |
| halfcheetah-medium-expert | $44.0 \pm 1.6$ | 62.4 | $106.3 \pm 1.9$ | $107.1 \pm 2.0$ | $\mathbf{114.0 \pm 1.6}$ | - |
| halfcheetah-medium-replay | $37.6 \pm 2.1$ | 46.2 | $61.3 \pm 1.9$ | $63.9 \pm 0.8$ | $\mathbf{69.7 \pm 1.4}$ | - |
| halfcheetah-full-replay | $62.9 \pm 0.8$ | *82.07* | $84.6 \pm 0.9$ | $84.5 \pm 1.2$ | $\mathbf{88.5 \pm 0.5}$ | $86.8 \pm 0.2$ |
| hopper-random | $3.7 \pm 0.6$ | 7.0 | $25.3 \pm 10.4$ | $31.3 \pm 0.0$ | $\mathbf{35.6 \pm 1.4}$ | - |
| hopper-medium | $54.1 \pm 3.8$ | 86.6 | $101.6 \pm 0.6$ | $100.3 \pm 0.3$ | $100.2 \pm 1.3$ | $\mathbf{103.8 \pm 0.1}$ |
| hopper-expert | $107.7 \pm 9.7$ | 109.9 | $110.1 \pm 0.1$ | $110.3 \pm 0.3$ | $\mathbf{112.0 \pm 0.2}$ | - |
| hopper-medium-expert | $53.9 \pm 4.7$ | 111.0 | $110.7 \pm 0.1$ | $110.1 \pm 0.3$ | $\mathbf{112.5 \pm 0.3}$ | $111.7 \pm 0.0$ |
| hopper-medium-replay | $16.6 \pm 4.8$ | 48.6 | $101.0 \pm 0.5$ | $101.8 \pm 0.5$ | $\mathbf{104.9 \pm 0.7}$ | - |
| hopper-full-replay | $19.9 \pm 12.9$ | *104.85* | $105.4 \pm 0.7$ | $102.9 \pm 0.3$ | $\mathbf{109.1 \pm 0.4}$ | - |
| Average | 49.9 | *70.9* | 85.2 | 84.5 | $\mathbf{89.6}$ | - |
| **Franka Kitchen** | **BC** | **SAC-Min** | **BEAR** | **IQL** | **CQL-Min** | **SPQR-CQL-Min** |
| kitchen-complete | 33.8 | 15.0 | 0.0 | $\mathbf{62.5}$ | 43.8 | $47.9 \pm 12.3$ |
| kitchen-partial | 33.8 | 0.0 | 13.1 | 46.3 | 49.8 | $\mathbf{54.2 \pm 7.2}$ |
| kitchen-mixed | 47.5 | 2.5 | 47.2 | 51.0 | 51.0 | $\mathbf{55.6 \pm 7.9}$ |
| Average | 38.4 | 5.8 | 20.1 | $\mathbf{53.3}$ | 48.2 | 52.6 |
| **Antmaze** | **BC** | **SAC-Min** | **BEAR** | **IQL** | **CQL-Min** | **SPQR-CQL-Min** |
| umaze | 65.0 | 0.0 | 73.0 | 87.5 | 74.0 | $\mathbf{93.3 \pm 4.7}$ |
| umaze-diverse | 55.0 | 0.0 | 61.0 | 62.2 | $\mathbf{84.0}$ | $80.0 \pm 0.0$ |
| medium-play | 0.0 | 0.0 | 0.0 | 71.2 | 61.2 | $\mathbf{80.0 \pm 8.2}$ |
| Average | 40.0 | 0.0 | 44.7 | 73.6 | 73.1 | $\mathbf{84.4}$ |

## E.2 Hypothesis Testing for Uniform Distribution

To check whether bias $e_{sa}^i$ follows a uniform distribution, it's sufficient to check $Q^i(s, a)$ follows a uniform distribution since $Q^*(s, a)$ is constant for fixed $s, a$. [27] propose a powerful test, called $\chi^2$ test which determines whether a given distribution follows uniform or not using hypothesis testing. We perform a $\chi^2$ test with significant level $\alpha = 0.025$ on 50 Q-ensemble trained in the D4RL hopper-random dataset in 3M timesteps with 10000 data of D4RL hopper-full-replay dataset. Since the $\chi^2$ test can be applied in a discrete random variable, we need to construct an approximative probabilistic mass function by binning. In Table 10, we show the acceptance rate of $\chi^2$ test over 10000 data with different bin sizes. Especially in experiments with 20 bins, SPQR-SAC-Min shows a better acceptance rate than SAC-Min approximately 40%.

Table 10: $\chi^2$ test

| Algorithm | bin=10 | bin=20 | bin=25 | bin=30 | bin=50 |
|---|---|---|---|---|---|
| SAC-Min | 20.08% | 32.12% | 36.38% | 38.54% | 50.99% |
| SPQR-SAC-Min | **33.67%** | **44.97%** | **49.78%** | **51.12%** | **63.56%** |

### E.3 Conservatism of ensemble and SPQR

We visualize how ensemble size $N$ affects the conservatism, and how SPQR can be computationally effective. As we described in Section 4.1 and Figure 3, SPQR has conservatism power. Increasing ensemble size $N$ for SAC-Min shows conservative behavior, as proven in [22] and Figure 9. We train SAC-Min for 3M timesteps in a halfcheetah-random dataset with ensemble size $N = 3, 5, 10, 15, 20, 50$. SAC-Min empirically shows conservative behavior as ensemble size increases. As a result, SPQR can be a computationally efficient algorithm when using SAC-Min.

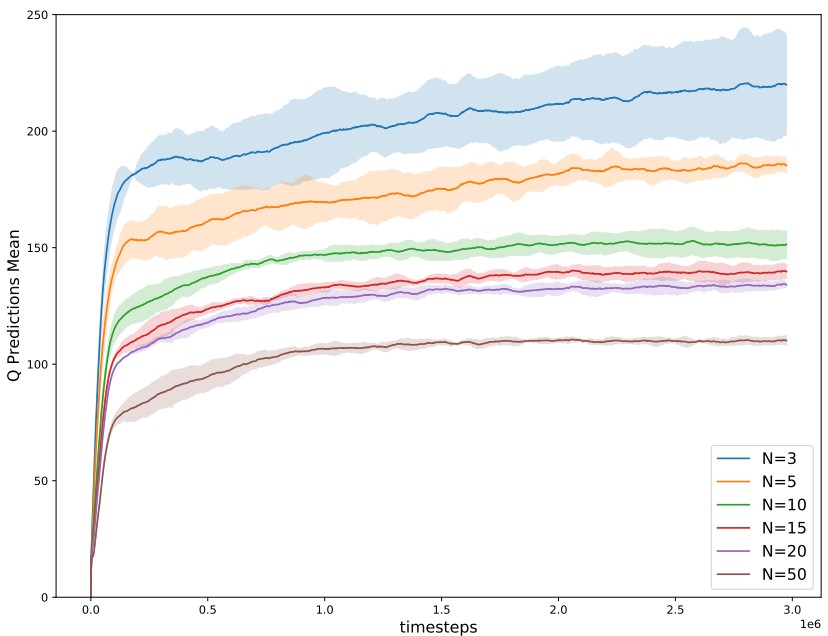

Figure 9: Mean of predicted Q-value with various $N$, averaged over 4 seeds

### E.4 Spike histogram visualization

We plot a more detailed visualization of the spike histogram for Figure 10. We visualize how the spike histogram of EDAC and SPQR varies compared with SAC-Min.

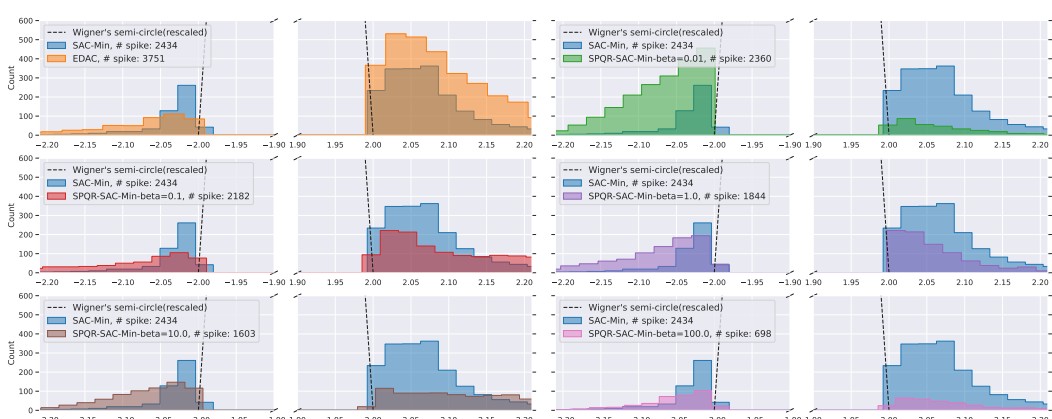

Figure 10: Spike visualization for SAC-Min, EDAC, and SPQR-$\beta = 0.01, 0.1, 1.0, 10.0, 100.0$.

### E.5 Bias of Q

In Figure 11, we plot the average normalized bias and standard deviation of normalized bias. Bias is calculated by Monte-Carlo estimation. SPQR-∗ decreases average bias and standard deviation of bias from baseline algorithms. For all environments, SPQR-SAC-Ens shows a smaller bias and standard deviation of bias than SAC-Ens. For Hopper and Humanoid tasks, SPQR-REDQ shows underestimation, and for Walker2d and Ant tasks, SPQR-REDQ shows unbiased estimation. For Ant, SPQR-REDQ shows less bias than REDQ.

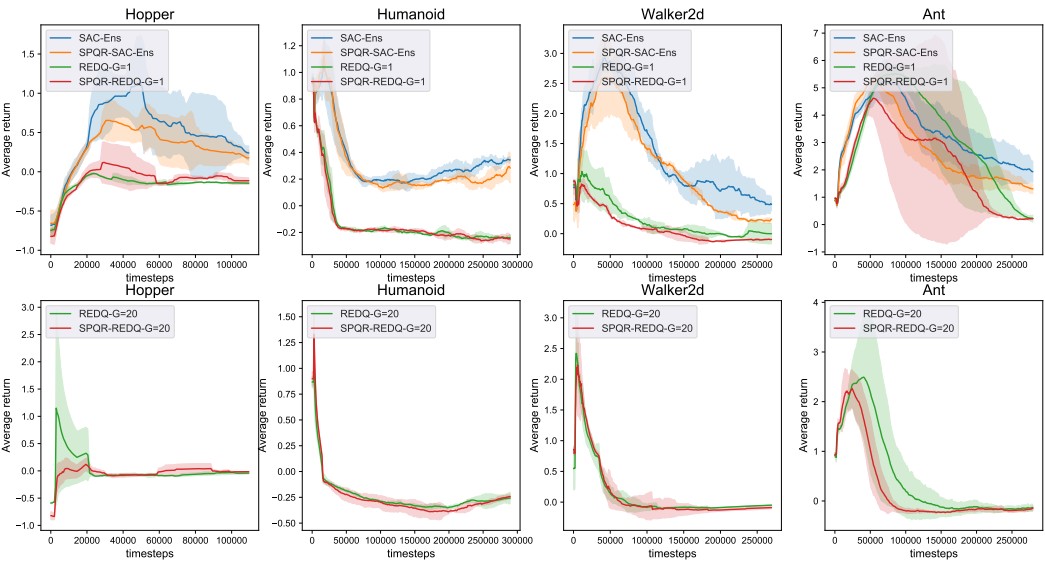

Figure 11: Average of bias and std of bias for SPQR-SAC-Ens and SPQR-REDQ, averaged over 4 seeds. G represents the UTD ratio.

### E.6 UTD Ablation Study

We visualize the performance of SPQR-SAC-Ens over variation of UTD ratio in Figure 12. SPQR-SAC-Ens shows performance improvement by increasing the UTD ratio with a conservative and stable bias.

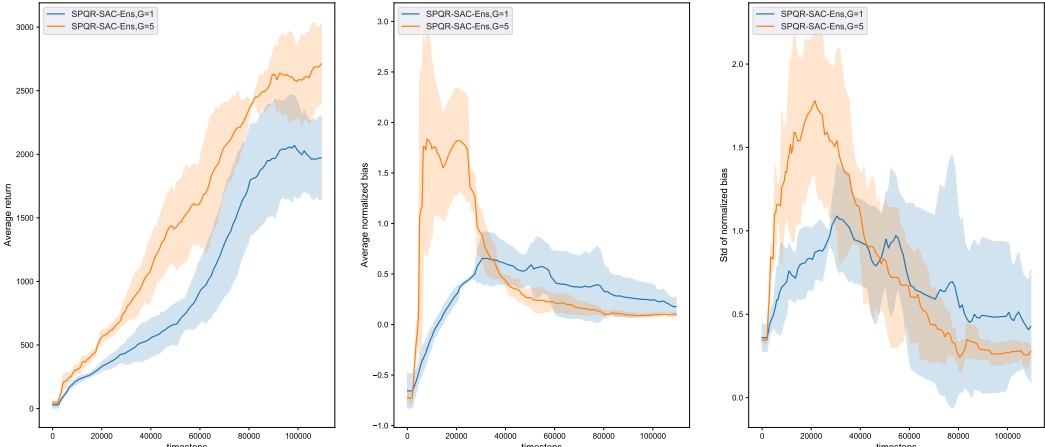

Figure 12: **Left 1:** Average return of SPQR-SAC-Ens for UTD ratio 1,5, **Left 2:** Average of bias of SPQR-SAC-Ens for UTD ratio 1,5, **Left 3:** Standard deviation of bias of SPQR-SAC-Ens for UTD ratio 1,5. All experiments are averaged over 4 seeds. G represents the UTD ratio.

### E.7 Early Collapse

To show how SPQR affects the ensemble when the agent collects new tuple $(s, a, r, s')$ in terms of exploration and variance, we evaluate the standard deviation with different $\beta$.

As [30] pointed out, the early collapse of Q-values occurs, which means that in an early stage of training, the variance of the Q-ensemble is almost zero since Q-values become similar to each other. To show how SPQR addresses the early collapse problem, we visualize the standard deviation of Q-ensemble in Figure 13. We train 10 Q-networks from the MuJoCo Hopper environment with the SAC-Min algorithm with the same hyperparameter. SPQR does not collapse Q-values during training, preserving appropriate variance of Q-value, and higher values of $\beta$ result in greater variance for $\beta = 0.3$ and $\beta = 2.0$.

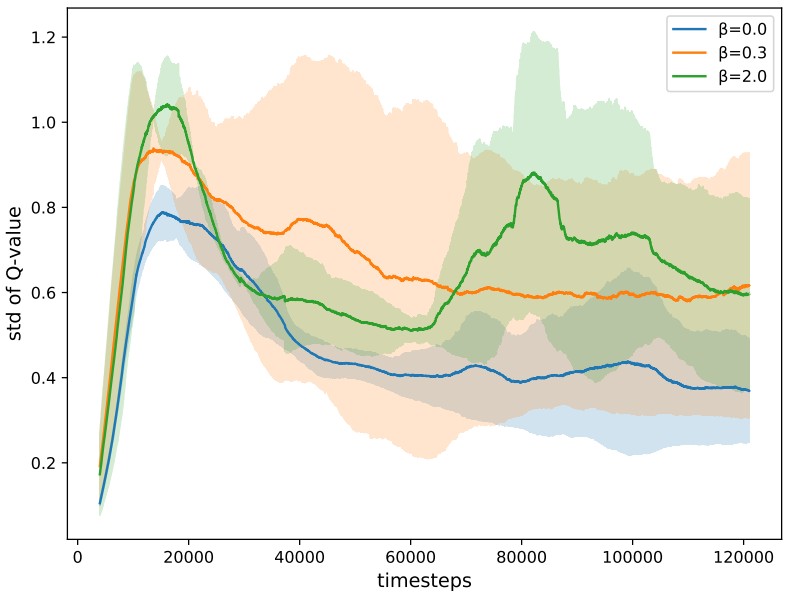

Figure 13: Standard deviation of Q-ensemble with different $\beta$, average over 3 seeds

### E.8 Hyperparameter Sensitivity

We evaluate SPQR over various $\beta$ to demonstrate hyperparameter sensitivity. We fix $N = 10$ for SPQR-SAC-Min and evaluate it on the D4RL Gym halfcheetah-random dataset and halfcheetah-medium dataset. Evaluation results are shown in Table 11.

Table 11: Performance of SPQR-SAC-Min over various $\beta$ on the D4RL halfcheetah-* dataset, averaged over 4 seeds

| dataset | $\beta = 0.0$ | $\beta = 0.01$ | $\beta = 0.1$ | $\beta = 0.5$ | $\beta = 1.0$ | $\beta = 2.0$ | $\beta = 5.0$ | $\beta = 10.0$ |
|---|---|---|---|---|---|---|---|---|
| random | $28.0 \pm 0.9$ | $30.6 \pm 3.4$ | $32.4 \pm 0.4$ | $30.6 \pm 1.3$ | $30.7 \pm 1.8$ | $31.2 \pm 2.2$ | $31.1 \pm 1.8$ | $32.3 \pm 2.4$ |
| medium | $67.5 \pm 1.2$ | $70.0 \pm 0.5$ | $71.7 \pm 0.6$ | $70.1 \pm 1.0$ | $70.5 \pm 1.7$ | $69.2 \pm 1.3$ | $70.0 \pm 0.9$ | $70.4 \pm 1.3$ |

## E.9 Compare with MED-RL

We implement and evaluate another diversifying method for online RL tasks, MED-RL. We use the Gini coefficient for MED-RL and use reported hyperparameters with UTD ratio 1. SPQR outperforms MED-RL for all tasks.

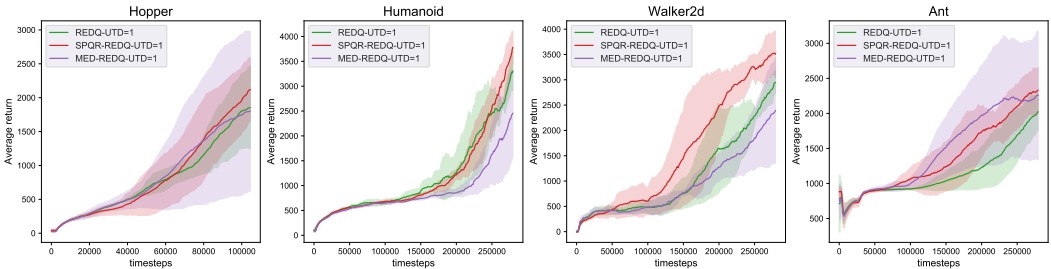

Figure 14: Average returns of REDQ, SPQR-REDQ and MED-RL with UTD ratio 1 on the MuJoCo environment, averaged over 4 random seeds.

## E.10 t-SNE Visualization

Figure 15 shows t-SNE [32] clustering visualization of Q-ensemble trained 3M timesteps by SPQR-SAC-Min algorithm at halfcheetah-medium-expert. We use t-SNE clustering for the last hidden layer representation of Q-ensemble with ensemble size 10 and 256 batch data at halfcheetah-medium-expert tasks. The t-SNE result shows that Q-ensemble trained by SPQR-SAC-Min is well clustered.

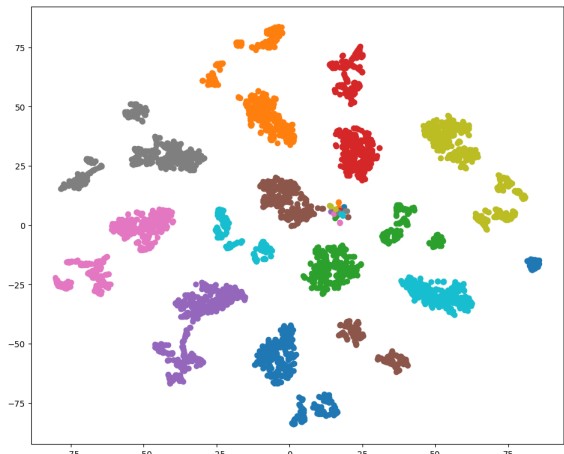

Figure 15: t-SNE for SPQR-SAC-Min at halfcheetah-medium-expert

### E.11 Correlation Visualization

We visualize a detailed figure for the Rightmost figure of Figure 1, the Pearson correlation coefficient matrix for a fully trained Q-ensemble and initial Q-ensemble in Figure 16. We train each Q-ensemble in the D4RL hopper-random dataset. For effective visualization, we sample randomly 10 Q-networks among 50 Q-networks and compute the Pearson correlation coefficient between each sampled Q-network with 256 batch data in a hopper-full-replay dataset. For the initial Q-ensemble, Q-networks are uncorrelated from each other. As we train Q-ensemble with SAC-Min for 3M timesteps, Q-networks become strongly correlated with each other.

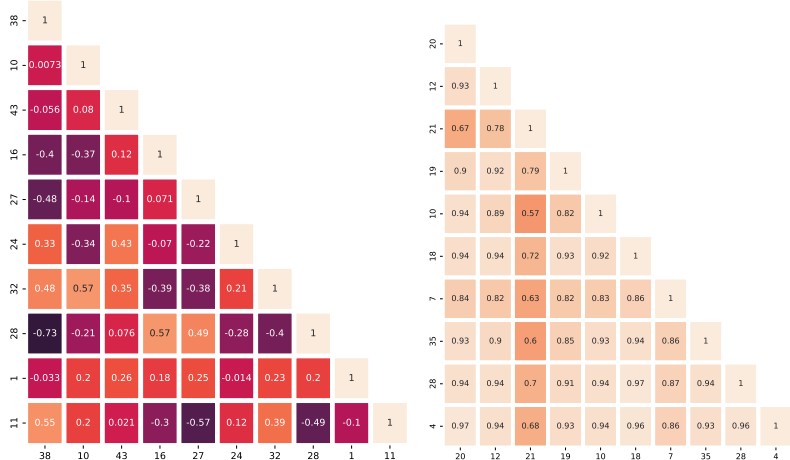

Figure 16: **Left:** Q-ensemble trained **0** timesteps. **Right:** Q-ensemble trained **3M** timesteps

