# OpenReview forum: "SPQR: Controlling Q-ensemble Independence with Spiked Random Model for Reinforcement Learning"
_NeurIPS.cc/2023/Conference — NeurIPS 2023 poster_

### Official Review · Reviewer_ppMG · 2023-07-03

**Soundness:** 3 good
**Presentation:** 2 fair
**Contribution:** 3 good
**Rating:** 6
**Confidence:** 2

**Summary:**

The paper proposes the SPQR loss to regularize the independence of the Q-network ensemble. The authors apply the random matrix theory and a spiked random model to derive the KL loss between Wigner’s semicircle distribution and the empirical spectral density of eigenvalues. The authors show that the independence could be enforced by minimizing this KL divergence loss with theoretical guarantees by satisfying the testing hypothesis. Extensive experiments on both the online and offline RL tasks and several baselines demonstrate that SPQR could improve the performance of the current ensemble RL algorithms promisingly by increasing of independence of the ensemble networks.

**Strengths:**

1.	This paper is well-organized and well-written.
2.	The proposed SPQR is with a simple format and shows a promising improvement over the ensemble RL methods such as SAC-ens and REDQ.
3.	The authors build the SPQR loss with mathematical tools and provide the theistical guarantee to ensure the independence of the ensembled Q-networks.
4.	Experiments on both the online and offline RL tasks are sufficient to validate the generality of the proposed SPQR and that it can improve the performance of ensemble RL methods.


**Weaknesses:**

1.	Detailed explanations between the used math tools and the proposed method are missing, making the understanding hard for the readers. For example, the connection between the random matrix theory and a spiked random model is not explained well. What is the purpose of the use of a spiked random model in SPQR?
2.	There lack details of in Algorithm 1. For example, how to construct the symmetric Q-matrix is not shown. The detailed version of the core algorithm is given in the appendix, which is strange.
3.	The quality of the figures needs to be improved. For example, the text size is too small.
4.	There are many typos.

a)	In Line 73, “tp” should be “to”.

b)	In Line 81, “SPQR analysis Q-ensemble independence …”.

c)	In Line 113, “Q-learning, We plot”.

d)	In Line 206, “Proof of Theorem 4.1 are …”.


**Questions:**

1.	What is the purpose of the use of spiked random model in SPQR? Could the authors provide some illustration examples or give more explanations?
2.	How to construct the symmetric Q-matrix in Algorithm 1?
3.	Why is the GOE chosen for the study and why should we relax the definition of GOE?


**Limitations:**

NA.

---

> ### Author Rebuttal · Authors · 2023-08-09
>
> We appreciate your considerate feedback and emphasis on improving our work. We address your questions and concerns as follows.
>
> Q1. Explanations about the spiked random model and GOE.
>
> A1.
>
> **1. spiked random model**
>
> The goal of the spiked random model is to detect/recover whether the given data is pure noise (random) or has a signal with noise (informative).
> As we visualize in Figure 2, the pure random matrix and informative matrix show different behavior in the eigenvalue domain.
>
> Before explaining the details, we can simply consider the spiked random model as the addition of an informative signal and the noise by the random matrix.
> In Figure 2, we generate a random matrix where the diversity of the elements in the 1st and the 3rd subfigure is similar but all elements of the random matrix in the 3rd figure are slightly moved towards (1,1)-direction.
> This difference is a clear difference between pure randomness and spiked randomness.
>
> Similar to PCA, the most natural assumption for the presence of information in the data is that the information can be captured in the most significant eigenvalue.
> To analyze the eigenvalue behavior of data, we need the random matrix theory.
> For example, in Right 3-4 in Figure 2, some signal, u,v exists and in the eigenvalue domain, it is detected by the top eigenvalue (rightmost) when the other eigenvalues follow Wigner’s semicircle law based on random matrix theory.
> On the other hand, in Left 1-2 in Figure 2, the elements of the random matrix show pure randomness without any signal, so its whole eigenvalues are in the Wigner’s semicircle.
>
> Our approach is to determine and alleviate non-independence among the Q-ensemble.
> Initially, we adapt the spike random model for SPQR to determine whether these data are collected independently.
> In order to make the Q-ensemble independent, we optimize the test criterion of the spiked random model.
> The test criterion has a form of KL loss between the eigenvalue distribution of observed data and pure random matrix, which is determined using random matrix theory.
>
> **2. GOE**
>
> In many machine learning and deep learning literature, GOE is used as the most natural way to model and analyze general random data matrix, such as analyzing loss surface [1], representation of GAN data [2], optimization acceleration [3], and explorations [4].
> We choose GOE to model the behavior of an independent ensemble.
>
> **3. relax the definition of GOE**
>
> Thank you for pointing out the sloppy statement.
> The phrase ‘relaxed definition’ on line 141 should have been written as ‘simplified terms’.
> For convenience, we intend to give a friendly definition to make it easier for GOE to understand.
> Actually, the type of distribution (Gaussian, Rademacher, Poisson, etc..) of entities does not affect the eigenvalue distribution (of course Wigner’s law) of the random matrix, which is called ‘universality’ in the random matrix theory literature. as we have noted in line 205.
>
> [1] Baskerville, Nicholas P., et al. "The loss surfaces of neural networks with general activation functions." Journal of Statistical Mechanics: Theory and Experiment 2021.6 (2021): 064001.
>
> [2] Seddik, Mohamed El Amine, et al. "Random matrix theory proves that deep learning representations of gan-data behave as gaussian mixtures." International Conference on Machine Learning. PMLR, 2020.
>
> [3] Lacotte, Jonathan, and Mert Pilanci. "Optimal randomized first-order methods for least-squares problems." International Conference on Machine Learning. PMLR, 2020.
>
> [4] Sagun, Levent, et al. "Explorations on high dimensional landscapes." arXiv preprint arXiv:1412.6615 (2014).
>
> Q2. Explanation about how to construct the symmetric Q-matrix.
>
> A2.
> It was really hard to include the full version of the algorithm because of the page limitation.  However, we agree with your comment.
> In the revised version, we will provide a more detailed explanation in the main text too.
>
> Our goal is to generate a symmetric Q-value matrix filled with an ensemble size of N.
> To avoid training or capturing the order of a matrix, we shuffle the filling order.
> This can be achieved by sorting the list of Q-networks l_k, as specified in Algorithm 3.
> In order to create a symmetric matrix of maximum size, the upper triangle part and the corresponding lower triangle part are equally filled.
> The size of the matrix is set as D=floor((sqrt(1+8*N)-1)/2), as described in Algorithm 3.
> D can be found by finding the largest integer that satisfies (D**2+D)/2 <= N.
>
> Please refer to the illustration in the **attached pdf files of the global response** for a clear understanding of the process.
>
> Q3. Typos, grammatical errors, and font size mistakes.
>
> A3.
> Thank you for kindly pointing out our typos, grammar, and font size mistakes.
> We will handle all of these mistakes in the revision.
>
> We hope that our explanation may address much of your concern.

---

> > ### Comment · Reviewer_ppMG · 2023-08-13
> >
> > The reviewer appreciate the authors' response. My concerns are effectively addressed and I would like to raise my score to 6.

---

> > > ### Author Response · Authors · 2023-08-13
> > > **Thanks**
> > >
> > > We appreciate your insightful and positive feedback. We will incorporate your helpful review into the revised version.

---

### Official Review · Reviewer_ve3n · 2023-07-04

**Soundness:** 4 excellent
**Presentation:** 3 good
**Contribution:** 4 excellent
**Rating:** 7
**Confidence:** 4

**Summary:**

The paper deals with the problem of alleviating overestimation bias in RL, using ensembles Q-functions. The authors argue that previous methods do not provide a theoretical guarantee of the independence of the members of the ensemble. To provide this they propose an approach based on random matrix theory, which, in practice, can be implemented as a regularization loss.

The approach is evaluated in a number of settings for RL in tasks in MuJoCo, D4RL Gym, Franka Kitchen and Antmaze.

**Strengths:**


+ The approach has a strong, non-trivial theoretical foundation.
+ Experimental results show a significant improvement over the baselines.

**Weaknesses:**

- Throughout the paper, there are many grammatical errors and awkward formulations. Some of these might hinder understanding, such as on page 5:
"high correlation occurs, which cannot be benefited by the ensemble method". I assume it this case what the authors mean is "benefit the ensemble method".
Many of these could be easily fixed with a grammar checker.

- The text on figures 5 and 6 (and to a lesser degree, figure 4) are unreadably small. Graphs should be redrawn in such a way that the text is still readable at a normal printing size.

**Questions:**

The meaning of Figure 3 is not clear. It seems that the higher the \beta, the lower the average Q value. But at this point in the paper, the approach had not yet been described, so it is not clear what the \beta controls. Also, we don't know the correct value of Q. I can make some assumptions that the figure aims to show how the proposed technique improves on the overestimation bias, but this (if true) would need to be explained more clear.

**Limitations:**

No separate limitations section found in the paper.

The theoretical nature of the paper does not raise ethical concerns.

---

> ### Author Rebuttal · Authors · 2023-08-09
>
> We appreciate the careful reading and interesting review. We address your questions and concerns as follows.
>
> Q1. Detailed explanation about Figure 3.
>
> A1.
> Thank you for providing a detailed and thoughtful comment to improve the presentation of our paper.
> During the rebuttal phase, it is not possible to modify our paper itself.
> Therefore, we provide more specific explanations for other readers as follows.
>
> To help to understand the independence and diversity of Q-ensemble more clearly, we mentioned in line 176 that beta can be simply considered as a loss gain (weight) and presented Figure 3 before providing a detailed explanation about SPQR and beta.
> We agree with your comment and will include the statement “beta is the loss weight for the regularization for Q-ensemble independence. Higher beta represents highly-likely-to-independent”  in the caption for Figure 3 in the revision.
>
> Section 4.1 aims to demonstrate the relationship between independence and conservatism is related, thereby enabling the control of conservatism through independence regularization.
> It does not directly address the alleviation of overestimation bias, which requires a true Q-value.
> The main purpose of this plot is to demonstrate the order of Q-values in terms of beta level, indicating that a higher independence regularization weight leads to a more conservative algorithm.
> The value and scale themselves are not our primary concerns in this section.
> Therefore, appropriate conservatism can be achieved by tuning beta, which prevents underestimation due to extreme conservatism.
>
> Q2. Typos, grammatical errors, and font size mistakes.
>
> A2.
> Thank you for kindly pointing out our typos, grammar, and font size mistakes.
> We will handle all of these mistakes in the revision.
>
> We hope that our explanation may address much of your concern.

---

> > ### Comment · Reviewer_ve3n · 2023-08-17
> > **Thank you for the rebuttal**
> >
> > Thank you for the rebuttal and the proposed changes in the paper.

---

> > > ### Author Response · Authors · 2023-08-18
> > > **Thanks**
> > >
> > > We appreciate your insightful and positive feedback. We will incorporate your helpful review into the revised version.

---

### Official Review · Reviewer_BeK9 · 2023-07-06

**Soundness:** 4 excellent
**Presentation:** 3 good
**Contribution:** 3 good
**Rating:** 7
**Confidence:** 4

**Summary:**

This paper proposes a new regularization loss that improves the independence of Q ensemble, thus improving the performance on online and offline DRL settings. The authors first point out that previous works with Q ensemble either rely on assumptions that are inaccurate in practice, or rely on heuristics to improve diversity of networks and no theoretical support. The authors provide theoretical results that support the proposed method, and also conducted empirical experiments, where the proposed method is applied to multiple recent online and offline algorithms, and tested on the MuJoCo online and d4rl offline benchmarks. On a high level, the regularization (Spiked Wishart Q-ensemble independence reuglarization (SPQR)) encourages the Q ensemble distribution to be closer to an ideal independent ensemble, resulting in a more diversified Q prediction values, lower bias and better performance.

**Strengths:**

**originality**
- The theoretical results and the proposed algorithm, the findings can be seen as novel
- Learning with Q ensembles has been studied in many previous works, but as the authors pointed out, they either rely on assumptions that are inaccurate in practice, or rely on heuristics, so the novelty is quite good here.

**quality**
- Overall the presentation is good, writing is clear

**Clarity**
- The motivation, connection to related works are all quite clear, and ample technical details are given, the authors also explained that they try to be fair in the comparisons and use the same hyperparameters. And the authors explained computation time, performance improvement, implementation difficulty quite clearly.

**significance**
- the analysis is nice, which shows that the proposed method seems to achieve the desired improved independence.
- empirical results showing consistent performance improvement over the baseline, big improvement in some settings, slight improvements in others, but overall quite consistent. Consistent improvement over recent sota algorithms such as REDQ with small change in the code and low computation cost and no excessive fine-tuning is quite impressive.
- the theoretical part is interesting and quite important

**Weaknesses:**

I don't have very major concerns, one thing that can be fixed is the text in all your figures, especially those in the main paper are just too small, please make the fontsize of legend, as well as axis labels and ticks and any other text in the figures bigger so they are easier to read.

Minor issue:
- line 81: SPQR analysis -> analyzes ?
- Figure 3 caption can you add a short sentence on what does beta do? It is not explained until a later section and I got confused when reading to Figure 3.

**Questions:**

- Main suggestion: fix the figures and make them more readable.
- Question: what is the limitation of your work? Are these cases where the proposed method might fail, or type of algorithms that it cannot be applied to?

**Limitations:**

The proposed method seems to be quite general and can be applied to other methods based on Q ensembles. However can be good to have some discussion on limitations in the paper.

---

> ### Author Rebuttal · Authors · 2023-08-09
>
> Thank you for your interest in our research and for providing us with constructive feedback.
> We address your questions and concerns as follows.
>
> Q1. Typos, grammatical errors, and font size mistakes.
>
> A1.
> Thank you for kindly pointing out our typos, grammar, and font size mistakes.
> We will take care of all the mistakes in the revision.
>
> Q2. More detailed caption for Figure 3.
>
> A2.
> Thank you for providing a detailed and thoughtful comment to improve the presentation of our paper.
> During the rebuttal phase, it is not possible to modify our paper itself.
> Therefore, we provide more specific explanations for other readers as follows.
>
> To help to understand the independence and diversity of Q-ensemble more clearly, we mentioned in line 176 that beta can be simply considered as a loss gain (weight) and presented Figure 3 before providing a detailed explanation about SPQR and beta.
> We agree with your comment and will include the statement “beta is the loss weight for the regularization for Q-ensemble independence. Higher beta represents highly-likely-to-independent”  in the caption for Figure 3 in the revision.
>
>
> Q3. Limitation of our work.
>
> A3.
> We might need a hyperparameter tuning on the regularization weight beta. However, we can broadly apply a common fixed beta in most environments since beta is not a sensitive hyperparameter as we have already shown in Table 8 and Table 11 in the appendix.
>
> We hope that our explanation may address much of your concern.

---

> > ### Comment · Reviewer_BeK9 · 2023-08-13
> > **Thank you for the rebuttal**
> >
> > I thank the authors for the rebuttal, I don't have other major concerns at this point.

---

> > > ### Author Response · Authors · 2023-08-14
> > > **Thanks**
> > >
> > > We appreciate your considerate and positive feedback. We will incorporate your helpful review into the revised version.

---

### Official Review · Reviewer_JH5V · 2023-07-07

**Soundness:** 3 good
**Presentation:** 3 good
**Contribution:** 2 fair
**Rating:** 5
**Confidence:** 3

**Summary:**

This work proposes a spiked Wishart Q-ensemble independence regularization (SPQR) to improve the independence of ensembling in Q-learning. SPQR encourages the ensemble to be closer to an ideal independent ensemble by penalizing the KL divergence between the eigenvalue distribution of the current ensemble and an ideal one.

**Strengths:**

The paper is easy to follow. The paper provides nice evidence of lack of independence in Q-ensemble training in current methods. The regularization to increase the diversity is well motivated, and seems simple and practical. The empirical evaluation is comprehensive, though more baselines would make them more compelling.

**Weaknesses:**

There is quite a bit of prior work on ensembling in deep RL, for example bootstrapped DQN [1] and MSG [2], which are not discussed. In particular, [2] provides theoretical and empirical support for constructing independent ensembles. A discussion and comparison with MSG is warranted given the emphasis on constructing independent ensembles.

While the emphasis of work is on improving ensembling methods in deep RL, recent methods have been able to forego ensembling while maintaining or improving performance over REDQ, for example DroQ [3] or RLPD [4]. It would be good to contextualize the improvements from SPQR by adding comparisons with these methods.

More comparisons with offline RL methods would help too, for example, IQL.

[1] Deep Exploration via Bootstrapped DQN. Osband et al.
[2] Why So Pessimistic? Estimating Uncertainties for Offline RL through Ensembles, and Why Their Independence Matters. Ghasemipour et al. NeurIPS, 2022.
[3] Dropout Q-Functions for Doubly Efficient Reinforcement Learning. Hiraoka et al.
[4] Efficient Online Reinforcement Learning with Offline Data. Ball et al.


**Questions:**

See weaknesses above.

**Limitations:**

While I do not see potential negative societal impact, a discussion on limitations is missing from the paper.

---

> ### Author Rebuttal · Authors · 2023-08-09
>
> We thank the reviewer for your detailed feedback. We address your questions and concerns as follows.
>
> Q1. Comparison with other RL algorithms.
>
> A1.
> We appreciate for informing us about meaningful prior studies. We will try our best to compare and evaluate each previous study based on their conceptual and empirical performance perspectives.
>
> 1. bootstrapped DQN
>
> Bootstrapped DQN aims to enhance deep exploration in the Atari environment by using a multi-head ensemble in deep online RL.
> SPQR aims to improve Q-ensemble independence in both online and offline RL.
> Bootstrapped DQN proposed a multi-head architecture for ensemble Q-learning, which was outlined and evaluated as multi-head MSG in the MSG paper.
> Intuitively, from the Q-ensemble independence perspective, separate N ensemble networks are preferable to a shared multi-head architecture.
> Moreover, empirical evidence indicates that deep ensemble MSG outperforms multi-head MSG.
> As bootstrapped DQN suggested, we will evaluate SPQR using a multi-head architecture in addition.
>
> 2. MSG
>
> The Model Standard-deviation Gradient (MSG), proposes a pessimistic ensemble Q-learning algorithm using the Upper Confidence Bound (UCB) of independent Q-networks, without relying on a shared target Q-value.
> SPQR aims to enhance independence by regularizing the network while maintaining a shared target Q-value.
> The performance of MSG and SPQR in the Antmaze environment can be compared, as reported in MSG.
> However, it should be noted that MSG uses the *-v0 environment, which needs to be reproduced.
> Since MSG and SPQR are orthogonal methodologies, we can combine them as we did for SPQR-EDAC.
> MSG claims that a shared target Q-value leads to optimism.
> We can modify the shared Q-target value of SPQR to an independent target Q-value or a multi-head target Q-value, as MSG has proposed.
>
> 3. DroQ
>
> The objective of DroQ is to enhance the computational efficiency of the ensemble Q-learning using the dropout technique.
> Since the performance of DroQ and REDQ is similar, we have thought that a comparison between REDQ and SPQR is sufficient.
> We will reproduce DroQ and evaluate an empirical comparison with SPQR for a revision of our paper.
>
> 4. RLPD
>
> The RLPD proposes an ensemble Q-learning algorithm that performs well by using an offline data buffer during online RL training.
> By combining RLPD with SPQR, performance can be improved as RLPD employs the orthogonal methodology of SPQR.
>
> 5. IQL
>
> Table 9 in Appendix E.1 compares SPQR-CQL with IQL in the Franka Kitchen and Antmaze environments.
> Since CQL outperforms IQL in the D4RL Gym environment, comparing SPQR with CQL suffices for the Gym environment.
>
> We hope that our explanation may address much of your concern.

---

> > ### Comment · Reviewer_JH5V · 2023-08-12
> > **Thanks for the response**
> >
> > I have read the rebuttal, and I will maintain my score.

---

> > > ### Author Response · Authors · 2023-08-13
> > > **Thanks**
> > >
> > > Thanks again for providing a helpful review. We will incorporate your feedback into the revised version.

---

### Official Review · Reviewer_KBA2 · 2023-07-07

**Soundness:** 3 good
**Presentation:** 3 good
**Contribution:** 3 good
**Rating:** 6
**Confidence:** 3

**Summary:**

Mitigating overestimation bias is crucial for deep reinforcement learning. Existing works about the ensemble techniques for Q-learning have been explored to leverage the diversity of multiple Q-functions. The authors argue that there has been no attempt to ensure ensemble independence from a theoretical standpoint. The authors introduce a regularization loss for Q-ensemble independence, based on random matrix theory, called Spiked Wishart Q-ensemble Independence Regularization (SPQR). The authors incorporate SPQR into online and offline ensemble Q-learning algorithms. Experimental results show that SPQR surpasses baseline algorithms in both online and offline RL benchmarks, demonstrating its effectiveness in addressing overestimation bias and improving performance.

**Strengths:**

- Significance and Originality: The problem and viewpoint that the paper studies - how to address overestimation bias/out-of-distribution error in RL is an important problem, and there have been a number of works investigating how to tackle this problem based on ensemble learning. However, most of the methods assume the bias follows a uniform and independent distribution, which may not hold in practice. This paper aims to improve previous ensemble methods from this perspective with a theoretical guarantee of improved Q-ensemble independence. The viewpoint from the random matrix theory seems novel, and the authors also propose a practical and tractable implementation for the method, which makes it possible to be applied in standard high-dimensional tasks.

- Quality: The authors have done a comprehensive experiments to evaluate the methods in different setupds, and the proposed SPQR methods provides improvements in different aspects.

- Clarity: The paper is also well-written and easy to follow.



**Weaknesses:**

- It is claimed in the paper that the assumption that most previous works rely on may not be true (the i.i.d. assumption about the bias). Could the authors demonstrate this effect is generally invalid in most of the environments? It would be better to convince the readers (since previous methods also have great performance although this assumption may not hold in practice).

- Results in Figure 5 about the validation for the independence analysis are interesting. Does this generally hold in different tasks?

- The improvement in standard D4RL locomotion tasks is somewhat marginal (4% improvement) considering increasing computation costs.

- The results in Figure 6 about the performance of SAC-Ens is very low. Could authors better explain this result?

**Questions:**

Please find my questions in the weakness section.

**Limitations:**

No.

---

> ### Author Rebuttal · Authors · 2023-08-09
>
> We are grateful for your constructive and thoughtful feedback. We address your questions and concerns as follows.
>
> Q1. Verification of the invalidation of the i.i.d assumption in various environments.
>
> A1.
> As you mentioned, we evaluate the acceptance ratio in various environments and datasets.
> According to the table below, SAC-Min and EDAC demonstrate a lower acceptance ratio than SPQR for various environments and offline datasets.
> We evaluate each algorithm on *-full-replay (OOD data) with chi-square hypothesis testing for independence, with a significant level \alpha=0.025.
> For example, an agent trained by hopper-random is evaluated with hopper-full-replay, as we have noted on line 245.
>
> Table: independence testing acceptance ratio per environment
> | algorithm | hopper-random | halfcheetah-random | walker2d-random |
> |-----------|---------------|--------------------|-----------------|
> | SAC-Min   | 30.4%         | 13.3%              | 34.0%           |
> | EDAC      | 30.4%         | 6.7%               | 0.0%            |
> | SPQR      | 80.4%         | 51.0%              | 60.0%           |
>
> Table: independence testing acceptance ratio per dataset
> | algorithm | halfcheetah-random | halfcheetah-medium | halfcheetah-expert |
> |-----------|--------------------|--------------------|--------------------|
> | SAC-Min   | 13.3%              | 23.3%              | 32.0%              |
> | SPQR      | 51.0%              | 70.0%              | 74.0%              |
>
>
> Q2. Explain computational cost and performance improvement.
>
> A2.
> The computational cost can be analyzed in terms of two aspects: memory usage and training time per epoch.
> From a memory usage perspective, we have less memory than SAC-Min because SPQR uses a smaller number of ensembles, N (we have provided the specific values of N in Table 8) by the conservative ensemble property of SPQR as shown in Figure 3 and Figure 8.
> This significantly reduces the training time per epoch since lower memory usage also has a significant impact on time consumption.
> As mentioned in line 294, SAC-Min requires 500 networks for some tasks in a hopper environment, whereas SPQR only needs 50, making it **significantly more computationally efficient**.
> Furthermore, computing the regularization loss does not significantly increase computational cost in terms of training time per epoch, by increasing **only 5% with the same ensemble size**, as mentioned in line 239.
>
> Table: average ensemble size decreasing rate per environment
> | environments    | walker2d | halfcheetah | hopper |
> |-----------------|----------|-------------|--------|
> | decreasing rate | 9.9% (33.3->30)    | 35% (10->6.5)        | 79% (350->75)   |
>
> In addition, building upon our previous comment in line 290, we note that the performance improvements are **significantly greater** in the case of low dataset quality.
> Although there was a remarkable improvement in the low-quality dataset (*-random), the average improvement rate seems somewhat marginal because SAC-Min has already shown good performance in the high-quality dataset, such as *-expert series.
> Therefore, we can conclude that our algorithm outperforms the baseline by a large margin, even if its average value appears to be small.
>
> As we provide the performance gain by dataset quality below, we can consistently confirm that our method shows a higher performance improvement ratio in the worse-quality dataset.
>
> Table: performance gain per dataset quality
> | random | medium | expert | medium-expert | medium-replay | full-replay |
> |--------|--------|--------|---------------|---------------|-------------|
> | 16%    | 7.0%   | 2.3%   | 3.2%          | 7.3%          | 3.3%        |
>
> In conclusion, we want to emphasize that applying SPQR becomes more computationally efficient since it needs fewer networks and shows a larger performance gain when the dataset quality is worse.
>
> Q3. Explain the performance of SAC-Ens.
>
> A3.
> When compared to other baseline ensemble methods, SAC-Ens appears to perform poorly.
> Given that SAC-Ens calculates an average Q-value among N networks, it is highly likely that its performance is similar to vanilla SAC that uses a single network (ensemble), as shown in Figure 4 (learning curve of DQN, DDQN, and Average DQN) and Figure 8 (sensitivity analysis of DQN, DDQN, and Average DQN) of the Maximin paper[1].
> We also note that the reported learning curve of SAC-Ens is close to the performance of vanilla SAC as shown in Figure 1 of the REDQ paper[2].
>
> More importantly, using the average target Q-value (SAC-Ens) underperforms the minimum target Q-value (REDQ(UTD=1), SAC-Min), as reported in Figure 3 and Figure 4 (learning curve of Average DQN and Maxmin DQN) of the Maximin paper[1].
> This occurs because using the average target Q-value suffers more from the overestimation bias harshly compared to using a minimum target Q-value, as illustrated in Figure 10 in the Appendix of our paper.
>
> [1] Lan, Qingfeng, et al. "Maxmin q-learning: Controlling the estimation bias of q-learning." arXiv preprint arXiv:2002.06487 (2020).
>
> [2] Chen, Xinyue, et al. "Randomized ensembled double q-learning: Learning fast without a model." arXiv preprint arXiv:2101.05982 (2021).
>
> We hope that our explanation and additional experiments may address much of your concern.

---

> > ### Comment · Reviewer_KBA2 · 2023-08-14
> >
> > Thank you for the detailed response. I find the additional results (the two tables) for Q1 very interesting, which also better supports the claim in the paper. I hope the authors will incorporate the discussion into the revision. I therefore raised the score to 6.

---

> > > ### Author Response · Authors · 2023-08-14
> > > **Thanks**
> > >
> > > We appreciate your insightful and positive feedback. We will incorporate your considerate review into the revised version.

---

### Author Rebuttal · Authors · 2023-08-09

For reviewer ppMG, we attach a pdf file for illustration about constructing a symmetric matrix here.

---

### Decision · Program_Chairs · 2023-09-21

**Decision:**

Accept (poster)

**Comment:**

This paper has been initially positively evaluated by reviewers, and the rebuttal helped to solve concerns about the empirical validation of the proposed method, which resulted in some reviewers increasing their scores. Overall, there is a clear consensus for accepting this paper.

I encourage the authors to address all the comments and to incorporate the recommended improvements in the final version.